# PULP MOTION: FRAMING-AWARE MULTIMODAL CAMERA AND HUMAN MOTION GENERATION

**Robin Courant**[1]   **Xi Wang**[1]   **David Loiseaux**[1,2]   **Marc Christie**[3]   **Vicky Kalogeiton**[1]

[1]LIX, École Polytechnique, CNRS, IPP    [2]Inria Saclay    [3]Inria, IRISA, Univ Rennes, CNRS

## ABSTRACT

Treating human motion and camera trajectory generation separately overlooks a core principle of cinematography: the tight interplay between actor performance and camera work in the screen space. In this paper, we are the first to cast this task as a text-conditioned joint generation, aiming to maintain consistent on-screen framing while producing two heterogeneous, yet intrinsically linked, modalities: human motion and camera trajectories. We propose a simple, model-agnostic framework that enforces multimodal coherence via an auxiliary modality: the on-screen framing induced by projecting human joints onto the camera. This on-screen framing provides a natural and effective bridge between modalities, promoting consistency and leading to more precise joint distribution. We first design a joint autoencoder that learns a shared latent space, together with a lightweight linear transform from the human and camera latents to a framing latent. We then introduce auxiliary sampling, which exploits this linear transform to steer generation toward a coherent framing modality. To support this task, we also introduce the PulpMotion dataset, a human-motion and camera-trajectory dataset with rich captions, and high-quality human motions. Extensive experiments across DiT- and MAR-based architectures show the generality and effectiveness of our method in generating on-frame coherent human-camera motions, while also achieving gains on textual alignment for both modalities. Our qualitative results yield more cinematographically meaningful framings setting the new state of the art for this task. Code, models and data are available in our project page.

## 1 INTRODUCTION

Cinematography is inherently a collaborative task, shaped by the joint relationship between the actor and the director. On the one hand, the director's camera seeks to frame the actors, adjusting to their movements to capture the desired performance on screen. On the other hand, the actor must also remain attentive to the presence of the camera, *e.g.* pausing at a marker until the camera arrives, before continuing a movement. Such motions are not spontaneous but rather intentional. These carefully crafted choices aim at enhancing the cinematic aesthetics. Balancing natural performance with visual framing between actors and cameras remains a central challenge in filmmaking.

Prior works have typically addressed only one side of this joint problem, treating them as standalone modalities: either human motion generation (Zhang et al., 2024; Tevet et al., 2023; Jiang et al., 2023) or camera trajectory generation (Jiang et al., 2024a; Courant et al., 2024; Zhang et al., 2025a), but never both simultaneously. In this work, we introduce the text-conditioned task of jointly generating human motion and camera trajectories. This task is challenging, as any mismatch between motion and camera may lead to poor framing, how the characters are positioned on screen, or even empty frames (*e.g.*, the subject moving out of view). The root problem of this joint generative task, referred to in computer vision as multimodal generation, is to produce high-quality outputs for each modality while maintaining multimodal coherence.

Multimodal generation has been widely studied in domains such as video–audio (Ruan et al., 2023; Hayakawa et al., 2025) and image–text (Li et al., 2025; Xu et al., 2023b). However, most approaches either rely solely on paired data to capture multimodal relationships (Xie et al., 2025; Li et al., 2025; Swerdlow et al., 2025), explicitly enforce correlations through architectural or algorithmic designs (Hu et al., 2023; Ruan et al., 2023; Xu et al., 2023b; Tang et al., 2023), or require training

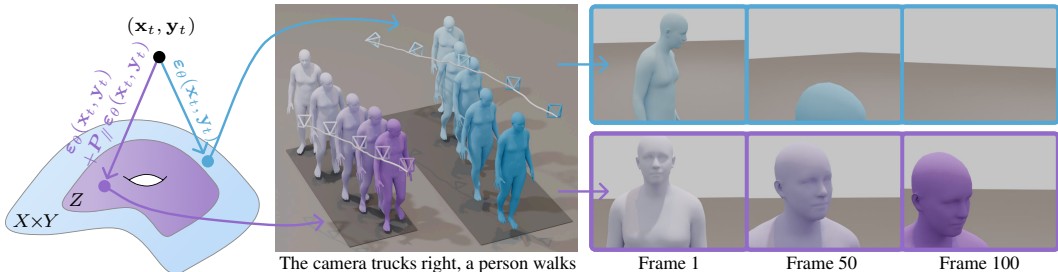

Figure 1: **Overview of our proposed auxiliary sampling.** We adapt the joint generation of $(\mathbf{x}, \mathbf{y})$ (camera trajectories and human motion) by leveraging an auxiliary modality $\mathbf{z}$ (on-screen human framing) to steer sampling toward more coherent joint generation via an orthogonal projection $\boldsymbol{P}_{/\!/}$. Specifically, our diffusion model predicts noise $\boldsymbol{\varepsilon}_\theta(\mathbf{x}, \mathbf{y})$, which is then adjusted along the auxiliary guidance direction.

adaptations or sampling guided by models trained on external data (Bao et al., 2023; Hayakawa et al., 2025; Xing et al., 2024; Kouzelis et al., 2025).

Training only on paired data provides an incomplete approximation of the joint data distribution (often due to mode coverage), making it challenging to sample precisely coherent modality pairs during generation. To address this, rather than adding architectural complexity as other methods do, we propose a multimodal generation framework that leverages an auxiliary modality as a bridge between generated modalities, steering the sampling process toward regions of higher multimodal coherence. Concretely, as illustrated in Figure 1, the model approximates an imperfect joint human–camera distribution (shown in blue). To mitigate this, we leverage the on-screen human framing within the camera as an auxiliary modality to steer the sampling toward a more coherent region of the joint distribution (shown in purple), *i.e.*, human and camera pair with a cinematic framing.

Our framework consists of two stages: (1) learning a joint latent space for human motion and camera trajectories, along with a linear transform which maps them into the auxiliary modality, *i.e.* the on-screen framing. This linear transform captures the relationship between the generated and auxiliary modalities directly in the latent space; (2) a sampling process augmented with an additional term derived from this linear transform, steering generation towards coherent multimodal generation.

For evaluation, we present PulpMotion, an extended version of a prior human-camera dataset, with more samples, motion captions, and higher-quality motion. We benchmark our approach on this dataset for both DiT-based (Peebles & Xie, 2023) and MAR-based (Li et al., 2024) architectures to demonstrate the generality and model-agnosticity of our approach. Our results show consistently improved coherence between generated motion and trajectories, yielding better framing quality and lower out-of-frame rates while preserving strong motion and trajectory generation performance.

Our contributions are: (1) a unified framework that jointly generates human motion and camera trajectories leveraging an auxiliary modality (on-screen framing) to enforce multimodal coherence during sampling, (2) the PulpMotion dataset, an extension of the prior human-camera dataset with more samples, motion captions, and higher-quality human motion, and (3) an extensive evaluation across multiple architectures demonstrating the method's generality and effectiveness.

## 2 RELATED WORK

**Human motion generation.** Diffusion-based approaches have driven recent progress (Ho et al., 2020; Rombach et al., 2022; Tevet et al., 2023; Kim et al., 2023; Zhang et al., 2024) on human motion generation, with extensions for efficient latent spaces, fast sampling, stronger textual alignment, and leverage of external data (Chen et al., 2023; Dai et al., 2024; Andreou et al., 2025; Zhang et al., 2023b). The newly proposed MAR architecture combines autoregressive and diffusion modeling and further pushes state-of-the-art performance (Li et al., 2024; Meng et al., 2024; Xiao et al., 2025). However, most methods treat motion as a *single modality*; although interactions with objects, people, and scenes are increasingly modeled (Peng et al., 2025b; Geng et al., 2025; Liang et al., 2024; Fan et al., 2024; Shan et al., 2024; Wang et al., 2024b; Cen et al., 2024), joint human–camera generation remains largely unexplored. Existing efforts typically use camera parameters only as constraints

or conditioning, rather than modeling their joint distribution with motion (Patel & Black, 2025; Ye et al., 2023; Wang et al., 2024a; Kocabas et al., 2024; Sun et al., 2023).

**Camera Trajectory Generation.** Camera control has evolved from handcrafted rules to learning-based methods that either mimic cinematic from example videos or optimize trajectories in differentiable 3D space (Blinn, 1988; Lino & Christie, 2015; Drucker et al., 1992; Jiang et al., 2020; 2021; Wang et al., 2023; Jiang et al., 2024c; Chen et al., 2024). To reduce reliance on exemplary data, Reinforcement Learning (RL)-based methods are often applied on drones and indoor scenes (Huang et al., 2019; Bonatti et al., 2020; Xie et al., 2023), but they remain environment-specific and style-limited. Diffusion-based camera generation, coupled with new datasets, further advances text-conditioned control and reduces reliance on curated reference videos (Jiang et al., 2024a; Courant et al., 2024; Wang et al., 2024e;d; Zhang et al., 2025a).
However, similarly to human motion generation, camera generation is also often regarded as a *single modality* problem conditioned on motion, rather than modeling the joint motion–camera distribution. In this work, we bridge this gap by adding human motion into the camera trajectory generation pipeline, modelling the synergy between how and what to film.

**Multimodal generation.** Most multimodal generation works leverage paired data to implicitly capture joint distribution, *e.g.*, text–image unified generation has been explored with different architectures: Dual Diffusion (Li et al., 2025) employs a DiT-based design, while Show-o (Xie et al., 2025) adopts an autoregressive backbone plus diffusion head framework. However, in practice, relying solely on paired data to learn implicit multimodal correlations often requires large datasets and still fails to fully capture multi-modal relationships.

Therefore, some works explicitly enforce multimodal coherence through architectural or algorithmic design. Hu et al. (2023) introduce a unified transition that compresses discrete representations across modalities under a discrete diffusion framework. MMDiffusion (Ruan et al., 2023) exploits a similar idea. It employs multimodal attention and random shifts to align multimodal information. Xu et al. (2023b) emphasizes architectural separation by disentangling context and data layers to encourage joint conditioning. Alternatively, CoDi (Tang et al., 2023) modifies cross-attention layers to emphasize a pre-aligned, modality-specific latent space, enabling any-to-any generation across multiple modalities. Despite their effectiveness in specific tasks, these approaches often depend on hand-crafted architectures, which limit their generality and adaptability across tasks and models.

Another line of work focuses on adapting only the training or sampling process, avoiding architectural modifications. For instance, UniDiffuser (Bao et al., 2023) trains a single multimodal diffusion with independent timesteps for each modality and applies an adapted classifier-free guidance scheme (CFG) (Ho & Salimans, 2021) with modality-specific timesteps. MMDisco (Hayakawa et al., 2025), inspired by classifier guidance (Dhariwal & Nichol, 2021), enables video–audio generation by training a joint discriminator to construct a guidance term during sampling, which is also used as regularisation for fine-tuning. Meanwhile, some works leverage foundation models to implicitly exploit larger datasets and stronger representations. For example, Xing et al. (2024) use the pre-trained multimodal binder ImageBind (Girdhar et al., 2023) to align generations via classifier-like guidance. Similarly, Kouzelis et al. (2025) design a representation-guidance term based on a diffusion generator trained on paired DINOv2 (Oquab et al., 2024) and image data, preserving the joint distribution without requiring an explicit classifier. However, these approaches still rely on adapting training or using large external pre-trained models, such as Imagebind or DINOv2.

In this work, we leverage an auxiliary modality to bridge the target modalities, steering sampling toward coherent joint generation in an architecture-agnostic manner, without requiring training adaptations or pre-trained models on external data. See extended discussion in Appendix B.

## 3 METHOD

**Problem formulation.** We consider a sample as a pair of human motion $\mathbf{x} \in \mathbb{R}^{N_\mathbf{x}}$ and camera trajectory $\mathbf{y} \in \mathbb{R}^{N_\mathbf{y}}$; both are sequences of $F$ frames. We aim to generate both modalities with respect to a textual description $\mathbf{c} \in \mathbb{R}^{N_\mathbf{c}}$, specifying the desired human motion and camera trajectory, *i.e.*, sampling from the joint human-camera distribution $p(\mathbf{x}, \mathbf{y}|\mathbf{c})$.

**Our approach.** Most related works capture multimodal relationships by relying exclusively on paired data, crafting specific architectural and algorithmic designs, introducing training adaptations

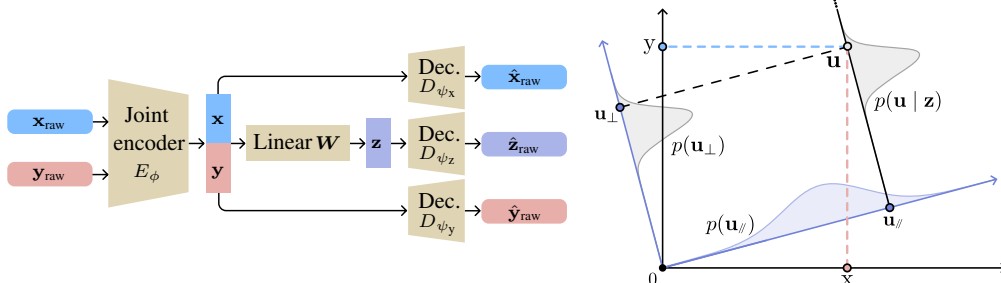

Figure 2: **Architecture of the multimodal autoencoder.** Human motion $\mathbf{x}_{\text{raw}}$ and camera trajectory $\mathbf{y}_{\text{raw}}$ are jointly encoded by $E_\phi$, linearly transformed via $\boldsymbol{W}$ into an auxiliary on-screen framing latent $\mathbf{z}$. Three decoders $D_{\psi_x}$, $D_{\psi_y}$, and $D_{\psi_z}$ reconstruct raw modalities: $\hat{\mathbf{x}}_{\text{raw}}, \hat{\mathbf{y}}_{\text{raw}}, \hat{\mathbf{z}}_{\text{raw}}$.

Figure 3: **Decomposition of $\mathbf{u} = [\mathbf{x}, \mathbf{y}]^\top$.** $\mathbf{u}$ decomposes onto two orthogonal components $\mathbf{u}_\perp$ and $\mathbf{u}_\parallel$. Our auxiliary sampling leverages this to encourage samples along $\mathbf{u}_\parallel$, parallel to the auxiliary modality $\mathbf{z}$.

or sampling strategies guided by external models. In contrast, our approach strengthens the coherence between modalities using an auxiliary modality, $\mathbf{z} \in \mathbb{R}^{N_\mathbf{z}}$ (with $N_\mathbf{z} < N_\mathbf{x} + N_\mathbf{y}$), which explicitly bridges them. In our setting, $\mathbf{z}$ represents the on-screen human framing, *i.e.*, the 2D projection of human joints in the camera view, a natural characteristic of the human-camera relationship.

Next, we describe our multimodal latent space with a latent linear transform of the auxiliary modality (Section 3.1) and present our auxiliary sampling scheme, which leverages the relationship between generated modalities and the auxiliary modality (Section 3.2).

## 3.1 MULTIMODAL LATENT SPACE

In multimodal generation, different modalities often exhibit varying properties, such as scale or geometric structure, which makes direct generation in the raw modality space challenging (Tang et al., 2023; Xing et al., 2024). Moreover, operating directly in the raw space increases computational and memory costs (Rombach et al., 2022) and can destabilize some diffusion losses (Meng et al., 2024).

To address these challenges, we adopt a latent diffusion framework for our joint human-camera generation task. Our latent representation is designed with two key aspects: (1) instead of embedding modalities separately, human and camera are aligned into a shared latent space, and (2) a lightweight learnable linear transform $\boldsymbol{W} \in \mathbb{R}^{(N_\mathbf{x}+N_\mathbf{y}) \times N_\mathbf{z}}$ that maps the latent human and camera representations into an on-screen framing latent, bridging both modalities.

More specifically, we propose an autoencoder architecture, shown in Figure 2. The model first maps the human $\mathbf{x}_{\text{raw}}$ and camera $\mathbf{y}_{\text{raw}}$ with a joint encoder $E_\phi$, producing latent embeddings $\mathbf{x}$ and $\mathbf{y}$. A learnable linear transform $\boldsymbol{W}$ then maps these embeddings into a on-screen framing latent $\mathbf{z}$:

$$\mathbf{z} = \boldsymbol{W} [\mathbf{x}, \mathbf{y}]^\top \quad . \tag{1}$$

Finally, three *independent* decoders $D_{\psi_x}$, $D_{\psi_y}$ and $D_{\psi_z}$ reconstruct each raw modality ($\mathbf{x}_{\text{raw}}, \mathbf{y}_{\text{raw}}, \mathbf{z}_{\text{raw}}$) from its respective latent[1]. The model is trained end-to-end with the following loss:

$$\mathcal{L}_{\text{AE}}(\phi, \psi_c, \psi_h, \psi_p) = \|D_{\psi_x}(E_\phi(\mathbf{x}_{\text{raw}}, \mathbf{y}_{\text{raw}})) - \mathbf{x}_{\text{raw}}\|^2 + \|D_{\psi_y}(E_\phi(\mathbf{x}_{\text{raw}}, \mathbf{y}_{\text{raw}})) - \mathbf{y}_{\text{raw}}\|^2$$
$$+ \|D_{\psi_z}(\boldsymbol{W} E_\phi(\mathbf{x}_{\text{raw}}, \mathbf{y}_{\text{raw}})) - \mathbf{z}_{\text{raw}}\|^2 \quad . \tag{2}$$

Note that the on-screen framing is never directly encoded; it is learned exclusively via the linear transform from the camera and human latents and supervised only through its reconstruction loss.

## 3.2 AUXILIARY SAMPLING

Given the multimodal latent space established in the previous section, we now introduce our multimodal latent diffusion framework, which incorporates an auxiliary sampling technique during inference to enhance multimodal coherence.

---

[1]Recall $\mathbf{z}_{\text{raw}}$ is defined as the 2D projection of human joints in the camera view, see Section 5.1 for details for $\mathbf{x}_{\text{raw}}, \mathbf{y}_{\text{raw}}, \mathbf{z}_{\text{raw}}$.

We train a generative model to produce multimodal representations of human motion $\mathbf{x}$ and camera trajectories $\mathbf{y}$ from textual descriptions c. For this, we adopt the standard Denoising Diffusion Probabilistic Model (DDPM) framework (Ho et al., 2020):

$$\mathcal{L}_{\text{noise}}(\theta) = \mathbb{E}_{t,\boldsymbol{\varepsilon}_{\mathbf{xy}}}\left[\|\boldsymbol{\varepsilon}_{\mathbf{xy}} - \boldsymbol{\varepsilon}_\theta\left(\mathbf{x}_t, \mathbf{y}_t, \mathbf{c}\right)\|^2\right] \quad, \tag{3}$$

where $\boldsymbol{\varepsilon}_\theta$ denotes the model's predicted noise corresponding to the true noise $\boldsymbol{\varepsilon}_{\mathbf{xy}}$ at timestep $t$.

**Auxiliary sampling.** Controllability in diffusion models is typically achieved via classifier-free guidance (CFG) (Ho & Salimans, 2021) over a conditioning signal $c$:

$$\begin{aligned}\nabla_{\mathbf{x}_t,\mathbf{y}_t} \log \tilde{p}(\mathbf{x}_t, \mathbf{y}_t|\mathbf{c}) &= \nabla_{\mathbf{x}_t,\mathbf{y}_t} \log p(\mathbf{x}_t, \mathbf{y}_t) \\ &+ w_c(\nabla_{\mathbf{x}_t,\mathbf{y}_t} \log p(\mathbf{x}_t, \mathbf{y}_t|\mathbf{c}) - \nabla_{\mathbf{x}_t,\mathbf{y}_t} \log p(\mathbf{x}_t, \mathbf{y}_t)) \quad.\end{aligned} \tag{4}$$

Note that the model's output is proportional to the score: $\boldsymbol{\varepsilon}_\theta(\mathbf{x}) \propto \nabla_{\mathbf{x}} \log p(\mathbf{x})$.

Here, CFG explicitly splits the score into an unconditional term and a conditional term, with the latter scaled by $w_c$. In our case, we aim to control the multimodal coherence between $\mathbf{x}$ and $\mathbf{y}$ through $\mathbf{z}$. Following the CFG strategy, we split the unconditional score term $\nabla_{\mathbf{x}_t,\mathbf{y}_t} \log p(\mathbf{x}_t, \mathbf{y}_t)$ in Equation (4) into a new "unconditional" term and an additional term in $\mathbf{z}$. To this end, we leverage the relationship in Equation (1) that links $\mathbf{z}$ to $\mathbf{x}$ and $\mathbf{y}$.

Let $\mathbf{u} = [\mathbf{x}, \mathbf{y}]^\top \in \mathbb{R}^{N_{\mathbf{x}}+N_{\mathbf{y}}}$. Since $\mathbf{z}$ is a compressed representation of $\mathbf{u}$ (*i.e.*, $N_{\mathbf{z}} < N_{\mathbf{x}} + N_{\mathbf{y}}$), it cannot fully capture all information in $\mathbf{u}$. We therefore decompose $\mathbf{u}$ into a $\mathbf{z}$-dependent component $\mathbf{u}_{/\!/}$ (*i.e.*, $\mathbf{u}_{/\!/} \mapsto \mathbf{z} = \boldsymbol{W}\mathbf{u}_{/\!/}$ is an isomorphism) and a complementary orthogonal component $\mathbf{u}_\perp$, such that $\mathbf{u} = \mathbf{u}_\perp + \mathbf{u}_{/\!/}$, as illustrated in Figure 3. This decomposition is precisely what we aim for: the component $\mathbf{u}_{/\!/}$ characterized by $\mathbf{z}$, steers the sampling toward a coherent $\mathbf{u}$, while the complementary component acts as an "unconditional" term.

**Lemma 3.1.** *Let $\boldsymbol{P}_{/\!/}$ denote the projection onto the orthogonal space of* $\ker(\boldsymbol{W})$. *Then, we have:*

$$\mathbf{u}_\perp := (\boldsymbol{I} - \boldsymbol{P}_{/\!/})\mathbf{u} \sim \mathcal{N}((\boldsymbol{I} - \boldsymbol{P}_{/\!/})\boldsymbol{\mu}, \sigma^2(\boldsymbol{I} - \boldsymbol{P}_{/\!/})) \quad and \quad \mathbf{u}_{/\!/} := \boldsymbol{P}_{/\!/}\mathbf{u} \sim \mathcal{N}(\boldsymbol{P}_{/\!/}\boldsymbol{\mu}, \sigma^2\boldsymbol{P}_{/\!/}) \quad, \tag{5}$$

*and the density of* $\mathbf{u}$ *decomposes as:*

$$p(\mathbf{u}) = p(\mathbf{u}_\perp)\, p(\mathbf{u}_{/\!/}). \tag{6}$$

Since $\boldsymbol{W}^\top \boldsymbol{W}$ is invertible in our setting, $\boldsymbol{P}_{/\!/}$ can be expressed as $\boldsymbol{P}_{/\!/} = \boldsymbol{W}(\boldsymbol{W}^\top \boldsymbol{W})^{-1}\boldsymbol{W}^\top$. See Figure 3 for illustration and Appendix C.2 for complete development and proof.

Thus, using Equation (6), the first term in Equation (4) can be split into an "unconditional" term over $(\mathbf{x}_t, \mathbf{y}_t)$ and a $\mathbf{z}$-dependent term weighted by $w_z$ that steers sampling toward $\mathbf{z}_t$:

$$\begin{aligned}\nabla_{\mathbf{x}_t,\mathbf{y}_t} \log \tilde{p}(\mathbf{x}_t, \mathbf{y}_t|\mathbf{c}) &= \nabla_{\mathbf{x}_t,\mathbf{y}_t} \log p(\mathbf{u}_\perp) + (1 + w_z)\nabla_{\mathbf{x}_t,\mathbf{y}_t} \log p(\mathbf{u}_{/\!/}) \\ &+ w_c(\nabla_{\mathbf{x}_t,\mathbf{y}_t} \log p(\mathbf{x}_t, \mathbf{y}_t|\mathbf{c}) - \nabla_{\mathbf{x}_t,\mathbf{y}_t} \log p(\mathbf{x}_t, \mathbf{y}_t)) \quad.\end{aligned} \tag{7}$$

Finally, we perform sampling using the following linear combination of the model's predictions, recalling that $\boldsymbol{\varepsilon}(\mathbf{x}) \propto \nabla_{\mathbf{x}} \log p(\mathbf{x})$, with more detailed derivation included in Appendix C.4:

$$\begin{aligned}\boldsymbol{\varepsilon}_\theta\left(\mathbf{x}_t, \mathbf{y}_t, \mathbf{c}, t\right) &= \boldsymbol{\varepsilon}_\theta\left(\mathbf{x}_t, \mathbf{y}_t, \emptyset, t\right) + w_z \boldsymbol{P}_{/\!/}\, \boldsymbol{\varepsilon}_\theta\left(\mathbf{x}_t, \mathbf{y}_t, \emptyset, t\right) \\ &+ w_c(\boldsymbol{\varepsilon}_\theta\left(\mathbf{x}_t, \mathbf{y}_t, \mathbf{c}, t\right) - \boldsymbol{\varepsilon}_\theta\left(\mathbf{x}_t, \mathbf{y}_t, \emptyset, t\right)) \quad.\end{aligned} \tag{8}$$

Since the final sampling does not explicitly condition on $\mathbf{z}_t$, there is no need to include the auxiliary modality during training. This reduces training cost and yields a more general approach.

## 4 PULPMOTION DATASET

Training a joint human–camera model requires paired data of human motions and camera trajectories. However, as shown in Table 1, most prior works provide only one modality, focusing either on human (*e.g.*, HumanML3D (Guo et al., 2022a)) or on camera (*e.g.*, RealEstate10k (Zhou et al., 2018)). More recently, the E.T. dataset (Courant et al., 2024) provides paired samples, but it prioritizes camera aspect, with lower-quality human motions and missing rich textual captions, making it inappropriate to train human motion models. This motivates us to introduce the PulpMotion dataset, a joint human-camera dataset with good-quality human motions along with motion captions.

We give an overview of PulpMotion in Section 4.1 and detail the extraction pipeline in Section 4.2.

Table 1: **Comparison of human and camera datasets.** We compare PulpMotion with existing human motion and/or camera trajectory datasets. We summarize modality coverage, available captions, dataset size (hours, frames, samples), sample length statistics (median, mean, std), and vocabulary size.

| Dataset | Camera | | Human | | #Hours | #Frames | #Samples | Sample lengths (frames) | | | #Vocabulary |
|---|---|---|---|---|---|---|---|---|---|---|---|
| | Traj | Caption | Motion | Caption | | | | Median | Mean | Std | |
| RealEstate10k (Zhou et al., 2018) | ✓ | ✗ | ✗ | ✗ | 121 | 11M | 79K | 115 | 136.9 | 80.0 | - |
| CamVid-30K (Zhao et al., 2024) | ✓ | ✗ | ✗ | ✗ | - | - | 30K | - | - | - | - |
| DynPose100k (Rockwell et al., 2025) | ✓ | ✗ | ✗ | ✗ | 157 | 6.8M | 100K | 63 | 67.97 | 17.91 | - |
| CameraBench (Lin et al., 2025) | ✗ | ✓ | ✗ | ✗ | - | - | 4K | - | - | - | - |
| CCD (Jiang et al., 2024a) | ✓ | ✓ | ✗ | ✗ | 50 | 4.5M | 25K | 189 | 180.4 | 69.6 | 48 |
| DataDoP (Zhang et al., 2025a) | ✓ | ✓ | ✗ | ✗ | 113 | 11M | 29K | - | 424.8 | - | 8,698 |
| KIT-ML (Plappert et al., 2016) | ✗ | ✗ | ✓ | ✓ | 12 | 0.8M | 4K | 71 | 99.0 | 99.6 | 1,623 |
| HumanML3D (Guo et al., 2022a) | ✗ | ✗ | ✓ | ✓ | 29 | 2M | 14K | 147 | 140.0 | 57.50 | 5,371 |
| Motion-X++ (Zhang et al., 2025b) | ✗ | ✗ | ✓ | ✓ | 181 | 19.5M | 120K | 152 | 167.9 | 125.33 | 8,116 |
| E.T. (Courant et al., 2024) | ✓ | ✓ | ✓ | ✗ | 120 | 11M | 115K | 75 | 93.9 | 73.8 | 1,790 |
| PulpMotion(Ours) | ✓ | ✓ | ✓ | ✓ | 314 | 22M | 193K | 107 | 117.3 | 63.6 | 4,599 |

Table 2: **Motion refinement and text-motion alignment.** We report metrics on the PulpMotion dataset, comparing raw extracted motions (Wang et al., 2024a) with our refined motions. Captions are generated either from human motions using m2t model (Jiang et al., 2023) or from RGB frames using our VLM-based approach (Bai et al., 2025).

| Motion | Caption | TMR-Score ↑ | R1 ↑ | R2 ↑ | R3 ↑ |
|---|---|---|---|---|---|
| Extracted | M2T | 4.08 | 1.16 | 2.47 | 3.63 |
| Extracted | VLM | 8.06 | 3.65 | 6.64 | 9.20 |
| Refined | M2T | 8.54 | 2.29 | 4.24 | 5.77 |
| Refined | VLM | **16.22** | **4.84** | **8.86** | **12.34** |

Table 3: **Motion refinement and motion quality.** We compare PulpMotion motion samples with HumanML3D (Guo et al., 2022a), evaluating raw extracted motions (Wang et al., 2024a) against our refined motions, using either m2t captions from human motions (Jiang et al., 2023) or our VLM-based captions from RGB frames (Bai et al., 2025).

| Motion | Caption | $FD_{TMR}$ ↓ | P ↑ | R ↑ | D ↑ | C ↑ |
|---|---|---|---|---|---|---|
| Extracted | - | 595.39 | 0.53 | 0.13 | 0.32 | 0.15 |
| Refined | M2T | 505.45 | 0.50 | 0.19 | 0.30 | 0.17 |
| Refined | VLM | **447.69** | **0.55** | **0.21** | **0.37** | **0.21** |

## 4.1 DATASET DESCRIPTION AND COMPARISON

Table 1 compares PulpMotion with existing human and camera datasets. Our dataset stands out by providing all modalities, camera trajectories and captions, human motions and captions, while most prior datasets cover only a subset. With 193K samples and 314 hours, PulpMotion is also the largest, nearly doubling the number of samples in E.T. (115K) and surpassing other motion-centric datasets such as Motion-X++ (120K). In terms of temporal coverage, PulpMotion exhibits longer sequences, with a median length of 107 frames, a mean of 117.3 frames, and a standard deviation of 63.6, indicating both richer and more diverse motion content compared to previous datasets.

## 4.2 EXTRACTION PIPELINE

**Human-camera pair extraction.** Following the E.T. extraction pipeline, we use TRAM (Wang et al., 2024a) to obtain 3D human-camera poses from videos of the CondensedMovies dataset (Bain et al., 2020), and apply the same post-processing steps (filtering, smoothing and cropping to a maximum length of 300 frames. For PulpMotion, we replace SLAHMR (Ye et al., 2023) with TRAM because it is significantly faster (∼1 fps vs. <0.1 fps), enabling large-scale processing. Moreover, TRAM provides higher-quality estimates, allowing us to keep trajectories that the E.T. pipeline previously filtered.

**Human-camera captions generation.** We generate detailed human motion captions inspired by HumanML3D (Guo et al., 2022a) using the Qwen2.5-VL (Bai et al., 2025) vision-language model, prompted with video clips and bounding boxes of the target character. The VLM follows annotation guidelines similar to HumanML3D. To assess the captioning quality we compute the motion-text alignment metrics (*i.e.* cosine similarity and retrieval recall) based on the TMR features (Petrovich et al., 2023). As shown in Table 2, our method achieves higher text-motion alignment than existing motion-to-text models (Jiang et al., 2023), attaining a TMR-Score of 8.06 against 4.08.

For camera captions, we follow the E.T. methodology: performing motion tagging and inputting it to a large language model (LLM) to produce user-friendly descriptions.

**Human motion refinement.** Outputs of TRAM often contain lower-quality human motion compared to mocap-based datasets like HumanML3D. To address this, we introduce a refinement step

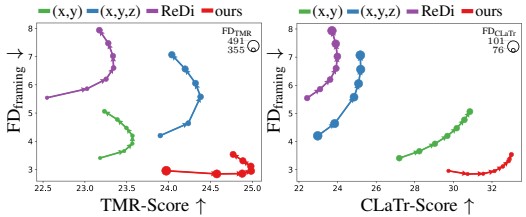

(a) FD$_{\text{framing}}$-TMR-Score    (b) FD$_{\text{framing}}$-CLaTr-Score

Figure 4: **Comparison in DiT on the mixed set.** Framing quality and modality-text alignment for $c$ guidance ranges from 5 to 11. The optimal region is at the bottom-right (low framing FD, high alignment).

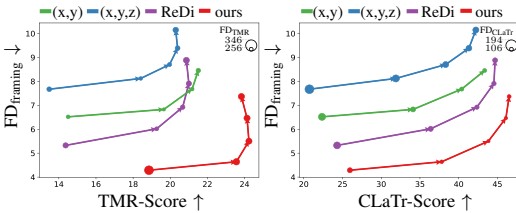

(a) FD$_{\text{framing}}$-TMR-Score    (b) FD$_{\text{framing}}$-CLaTr-Score

Figure 5: **Comparison in MAR on the mixed set.** Framing quality and modality-text alignment for $c$ guidance ranges from 1 to 5. The optimal region is at the bottom-right (low framing FD, high alignment).

to enhance motion quality. The main source of error in TRAM arises from partial observations; e.g., close-up shots capturing only the upper body. Therefore, to improve overall motion quality, we detect out-of-frame body parts via camera reprojection and refine these regions using the RePaint editing method (Lugmayr et al., 2022) with a HumanML3D-pretrained diffusion model. To assess the captioning quality we compute the motion quality metrics (*i.e.* Fréchet distance and PRDC (Naeem et al., 2020)) based on the TMR features (Petrovich et al., 2023). As shown in Table 3, this step significantly reduces the FD$_{\text{TMR}}$ score from 595.39 to 447.69.

We provide additional details on the dataset extraction pipeline in the Appendix D.1.

## 5 EXPERIMENTS

### 5.1 EXPERIMENTAL SETUP

**Data representation.**

**Framing features ($z_{\text{raw}}$).** We use the 2D Normalized Device Coordinates (NDC): screen-projected coordinates normalized to the range $[-1, 1]$, of nine key human joints (ankles, pelvis, spine, head, shoulders, and wrists). For a sequence of $F$ frames, this yields $X_{\text{framing}} \in \mathbb{R}^{F \times 18}$.

**Human features ($x_{\text{raw}}$).** We use the features introduced in Petrovich et al. (2024), for a motion of $F$ frames: $X_{\text{human}} = (r_z, \dot{r}_x, \dot{r}_y, \dot{\alpha}, \Theta, J) \in \mathbb{R}^{F \times 199}$ where $r_z \in \mathbb{R}^F$ is the Z (up) coordinate of the pelvis, $\dot{r}_x \in \mathbb{R}^F$ and $\dot{r}_y \in \mathbb{R}^F$ are the linear velocities of the pelvis, $\dot{\alpha} \in \mathbb{R}^F$ is the angular velocity of the Z angle of the body, $\Theta \in \mathbb{R}^{F \times 132}$ are the 22 first SMPL (Loper et al., 2023) pose parameters (6D representation (Zhou et al., 2019)), and $J \in \mathbb{R}^{F \times 63}$ are the 22 joints positions (pelvis excluded).

**Camera feature ($y_{\text{raw}}$).** We extend the features from Courant et al. (2024) by notably adding the intrinsics. For a trajectory of $F$ frames: $X_{\text{cam}} = (R, \dot{T}, D, F) \in \mathbb{R}^{F \times 14}$ where $R \in \mathbb{R}^{F \times 6}$ denotes rotation using the 6D continuous representation (Zhou et al., 2019), $\dot{T} \in \mathbb{R}^{F \times 3}$ is the linear velocity, $D \in \mathbb{R}^{F \times 3}$ is the relative distance bewteen the camera and the human, and $F \in \mathbb{R}^{F \times 2}$ encodes the horizontal and vertical fields of view (assuming the principal point lies at the image center).

**Metrics.**

**Framing metrics.** Since no existing metrics assess framing quality, we propose two metrics based on the 9-joint NDC representation introduced above. First, the Fréchet distance FD$_{\text{framing}}$ measures how well the on-screen framing of the generated camera and human matches a reference distribution. Second, the *Out-rate* is the fraction of frames where none of the 9 joints appear on-screen.

**Human metrics.** We use the standard text-to-motion metrics (Guo et al., 2020) using the TMR (Petrovich et al., 2023) feature space. We then report FD$_{\text{TMR}}$ and TMR-Score, and TMR-based R-precision. In addition, to evaluate how well generated samples span the variety of real data we compute the TMR-based coverage (Naeem et al., 2020).

**Camera metrics.** We use the metrics introduced in Courant et al. (2024). To evaluate the camera trajectory quality, we report the FD$_{\text{CLaTr}}$ and CLaTr-based coverage (Naeem et al., 2020); to evaluate the camera trajectory coherence, we report the CLaTr-Score and segmentation F1.

Table 4: **State-of-the-art comparison on the mixed subset.** We compare five baselines: human-conditioned camera generation $(\mathbf{x})$+DIRECTOR Courant et al. (2024), independent modality generation $(\mathbf{x})(\mathbf{y})$, dual-modality generation $(\mathbf{x}, \mathbf{y})$, triplet-modality generation $(\mathbf{x}, \mathbf{y}, \mathbf{z})$, and ReDi (Kouzelis et al., 2025), along with our auxiliary sampling (Aux). Results are reported for DiT (Peebles & Xie, 2023) and MAR (Li et al., 2024). Superscript $\pm$ denotes the 95% confidence interval over 10 samplings.

| Methods | Framing | | Human | | | | Camera | | | |
|---|---|---|---|---|---|---|---|---|---|---|
| | $FD_{framing}$ ↓ | Out-rate ↓ | $FD_{TMR}$ ↓ | TMR-Score ↑ | R3 ↑ | Coverage ↑ | $FD_{CLaTr}$ ↓ | CLaTr-Score ↑ | F1 ↑ | Coverage ↑ |
| Ground-truth | 0.00 | 0.89 | 0.00 | 17.72 | 22.00 | 1.00 | 0.00 | 68.88 | 87.43 | 1.00 |
| Auto-encoder | 0.23 | 4.61 | 124.78 | 18.16 | 21.81 | 85.30 | 15.64 | 57.98 | 67.04 | 86.64 |
| **DiT** | | | | | | | | | | |
| $(\mathbf{x})$+DIRECTOR | $22.21^{\pm0.03}$ | $60.56^{\pm0.14}$ | - | - | - | - | $95.46^{\pm0.82}$ | $24.44^{\pm0.15}$ | $27.00^{\pm0.14}$ | $50.50^{\pm0.29}$ |
| $(\mathbf{x})(\mathbf{y})$ | $11.21^{\pm0.12}$ | $48.02^{\pm0.24}$ | $357.99^{\pm0.52}$ | $25.03^{\pm0.06}$ | $4.34^{\pm0.12}$ | $10.55^{\pm0.17}$ | $67.76^{\pm0.20}$ | $46.74^{\pm0.11}$ | $46.71^{\pm0.24}$ | $53.66^{\pm0.36}$ |
| $(\mathbf{x})(\mathbf{y})$+Aux (ours) | $8.24^{\pm0.07}$ | $41.24^{\pm0.24}$ | $422.45^{\pm0.78}$ | $26.46^{\pm0.07}$ | $4.64^{\pm0.10}$ | $9.08^{\pm0.15}$ | $56.41^{\pm0.32}$ | $50.69^{\pm0.10}$ | $50.72^{\pm0.14}$ | $51.39^{\pm0.22}$ |
| $(\mathbf{x}, \mathbf{y})$ | $4.90^{\pm0.05}$ | $25.98^{\pm0.24}$ | $372.61^{\pm0.90}$ | $23.50^{\pm0.07}$ | $3.67^{\pm0.08}$ | $10.72^{\pm0.15}$ | $87.07^{\pm0.87}$ | $30.75^{\pm0.17}$ | $34.28^{\pm0.27}$ | $51.62^{\pm0.40}$ |
| $(\mathbf{x}, \mathbf{y}, \mathbf{z})$ | $4.18^{\pm0.03}$ | $23.88^{\pm0.19}$ | $390.08^{\pm1.20}$ | $23.88^{\pm0.12}$ | $3.22^{\pm0.11}$ | $11.58^{\pm0.13}$ | $97.45^{\pm0.61}$ | $23.34^{\pm0.16}$ | $27.40^{\pm0.18}$ | $50.80^{\pm0.44}$ |
| $(\mathbf{x}, \mathbf{y}, \mathbf{z})$+Aux (ours) | $3.76^{\pm0.03}$ | $13.90^{\pm0.22}$ | $532.42^{\pm1.00}$ | $24.58^{\pm0.05}$ | $6.13^{\pm0.12}$ | $6.88^{\pm0.18}$ | $106.97^{\pm1.03}$ | $24.61^{\pm0.20}$ | $27.43^{\pm0.21}$ | $43.36^{\pm0.34}$ |
| ReDi | $5.57^{\pm0.04}$ | $28.99^{\pm0.22}$ | $360.07^{\pm1.26}$ | $22.48^{\pm0.06}$ | $5.68^{\pm0.18}$ | $12.83^{\pm0.16}$ | $83.66^{\pm1.05}$ | $22.73^{\pm0.22}$ | $26.53^{\pm0.20}$ | $55.24^{\pm0.41}$ |
| $(\mathbf{x}, \mathbf{y})$+Aux (ours) | $3.37^{\pm0.02}$ | $16.76^{\pm0.19}$ | $431.54^{\pm1.15}$ | $25.05^{\pm0.07}$ | $3.89^{\pm0.14}$ | $8.91^{\pm0.13}$ | $80.08^{\pm0.76}$ | $32.81^{\pm0.19}$ | $36.06^{\pm0.25}$ | $48.68^{\pm0.20}$ |
| **MAR** | | | | | | | | | | |
| $(\mathbf{x})(\mathbf{y})$ | $11.59^{\pm0.08}$ | $51.05^{\pm0.24}$ | $296.01^{\pm0.73}$ | $21.71^{\pm0.09}$ | $11.69^{\pm0.12}$ | $17.48^{\pm0.23}$ | $111.42^{\pm0.75}$ | $51.96^{\pm0.11}$ | $51.69^{\pm0.11}$ | $49.85^{\pm0.41}$ |
| $(\mathbf{x})(\mathbf{y})$+Aux (ours) | $9.13^{\pm0.07}$ | $47.30^{\pm0.22}$ | $308.90^{\pm0.65}$ | $24.12^{\pm0.08}$ | $12.07^{\pm0.16}$ | $14.64^{\pm0.19}$ | $91.85^{\pm0.69}$ | $55.75^{\pm0.10}$ | $54.78^{\pm0.18}$ | $48.67^{\pm0.21}$ |
| $(\mathbf{x}, \mathbf{y})$ | $8.51^{\pm0.07}$ | $40.75^{\pm0.28}$ | $275.30^{\pm0.55}$ | $21.68^{\pm0.06}$ | $10.60^{\pm0.19}$ | $17.10^{\pm0.28}$ | $117.77^{\pm0.63}$ | $42.84^{\pm0.14}$ | $42.69^{\pm0.23}$ | $54.89^{\pm0.37}$ |
| $(\mathbf{x}, \mathbf{y}, \mathbf{z})$ | $8.66^{\pm0.09}$ | $37.50^{\pm0.17}$ | $268.41^{\pm0.71}$ | $20.13^{\pm0.08}$ | $10.59^{\pm0.11}$ | $19.83^{\pm0.33}$ | $148.12^{\pm0.96}$ | $38.58^{\pm0.10}$ | $38.34^{\pm0.09}$ | $51.74^{\pm0.41}$ |
| $(\mathbf{x}, \mathbf{y}, \mathbf{z})$+Aux (ours) | $6.48^{\pm0.07}$ | $30.19^{\pm0.26}$ | $288.23^{\pm0.84}$ | $22.71^{\pm0.08}$ | $11.27^{\pm0.11}$ | $16.26^{\pm0.20}$ | $143.10^{\pm1.01}$ | $41.03^{\pm0.16}$ | $40.71^{\pm0.17}$ | $49.68^{\pm0.35}$ |
| ReDi | $6.96^{\pm0.07}$ | $32.25^{\pm0.18}$ | $275.58^{\pm0.66}$ | $20.84^{\pm0.08}$ | $10.90^{\pm0.11}$ | $18.41^{\pm0.32}$ | $122.40^{\pm0.77}$ | $42.60^{\pm0.15}$ | $42.70^{\pm0.21}$ | $54.96^{\pm0.51}$ |
| $(\mathbf{x}, \mathbf{y})$+Aux (ours) | $6.42^{\pm0.04}$ | $33.65^{\pm0.23}$ | $301.39^{\pm0.25}$ | $24.46^{\pm0.07}$ | $11.28^{\pm0.09}$ | $14.14^{\pm0.14}$ | $108.74^{\pm0.46}$ | $45.96^{\pm0.14}$ | $45.39^{\pm0.22}$ | $53.67^{\pm0.38}$ |

## 5.2 COMPARISON TO THE STATE OF THE ART

In this section, we compare our **auxiliary sampling** (Aux) method against five baselines: (1) **human-conditioned camera generation** $(\mathbf{x})$+DIRECTOR Courant et al. (2024): a two-stage approach in which one model generates human motions, and a second model generates camera trajectories conditioned on those motions; (2) **independent modality generation** $(\mathbf{x})(\mathbf{y})$: two separate models for each modality; (3) **dual-modality generation** $(\mathbf{x}, \mathbf{y})$: a single model generates both modalities without the auxiliary modality; (4) **triplet-modality generation** $(\mathbf{x}, \mathbf{y}, \mathbf{z})$: a single model generates both modalities and the auxiliary modality; and (5) **ReDi** (Kouzelis et al., 2025): a single model for both modalities and the auxiliary modality, with representation sampling leveraging the auxiliary modality. Except for $(\mathbf{x})$+DIRECTOR (originally trained with DiT), we evaluate all baselines and our method using both DiT-based (Peebles & Xie, 2023) and MAR (Li et al., 2024) architectures (see Appendix E.2.1 for more details on architectures).

**Quantitative results.** Table 4 reports a comparison of our auxiliary sampling (Aux) method against state-of-the-art baselines across both DiT and MAR architectures on the mixed subset. We summarise our experimental observations as follows:

(i) **Auxiliary sampling consistently improves framing (multimodal coherence).** Applying Aux leads to systematic improvements in framing quality ($FD_{framing}$) and out-of-frame (Out-rate) rates across all baseline configurations and architectures. For DiT, auxiliary sampling reduces $FD_{framing}$ from $11.21 \rightarrow 8.24$ for $(\mathbf{x})(\mathbf{y})$, from $4.90 \rightarrow 3.37$ when applied to $(\mathbf{x}, \mathbf{y})$. A similar trend holds for MAR, where $FD_{framing}$ decreases from $11.59 \rightarrow 9.13$ for $(\mathbf{x})(\mathbf{y})$ and from $8.51 \rightarrow 6.42$ for $(\mathbf{x}, \mathbf{y})$. Out rates follow the same pattern: for DiT, Aux reduces the out rate from $48.02\% \rightarrow 41.24\%$ $((\mathbf{x})(\mathbf{y}))$ and from $25.98\% \rightarrow 16.76\%$ $((\mathbf{x}, \mathbf{y}))$, while for MAR it decreases from $51.05\% \rightarrow 47.30\%$ and $40.75\% \rightarrow 33.65\%$, respectively. Overall, auxiliary sampling achieves the best $FD_{framing}$ and out rates across both architectures (DiT and MAR): showing that it is architecture-agnostic and consistently enhances framing quality, i.e. multimodal coherence, for all settings (independent and joint).

(ii) **Conditioning on human motion alone is insufficient for strong framing.** We next examine the human-conditioned camera generation baseline $(\mathbf{x})$+DIRECTOR, where human trajectories are first generated using the same backbone as the independent setting $((\mathbf{x})(\mathbf{y}))$, and camera motion is subsequently conditioned on these synthesized humans. While this strategy is weaker than other baselines: for example, under DiT, $(\mathbf{x})$+DIRECTOR yields a $FD_{framing}$ of 22.21, compared to 8.24 for $(\mathbf{x})(\mathbf{y})$+Aux and 3.37 for $(\mathbf{x}, \mathbf{y})$+Aux. Its out rate is also higher with $60.56\%$ against $41.24\%$ and $16.76\%$, respectively. These results that conditioning on generated human motion alone provides limited contextual information for accurate camera framing. In contrast, joint generation methods, especially when combined with Aux, consistently outperform $(\mathbf{x})$+DIRECTOR and the independent baseline $((\mathbf{x})(\mathbf{y}))$, highlighting the importance of jointly modeling modalities during sampling.

(iii) **Auxiliary sampling strengthens per-modality alignment while preserving quality.** Beyond

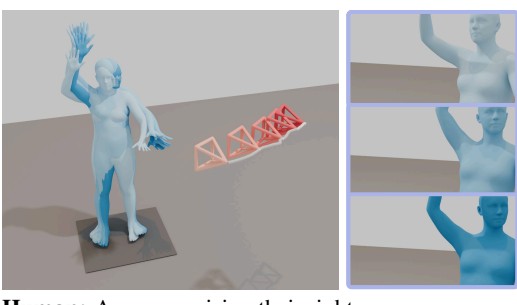
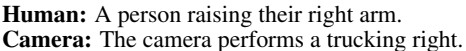
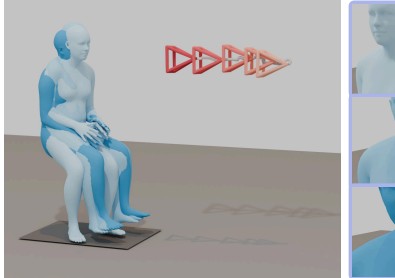

**Human:** A person raising their right arm.
**Camera:** The camera performs a trucking right.

Figure 6: **Example with DiT on the mixed subset.**

**Human:** A person sitting.
**Camera:** The camera performs a push in.

Figure 7: **Example with MAR on the mixed subset.**

framing, Aux improves text–modality alignment across both human and camera dimensions. For human motion, auxiliary sampling increases TMR-Score from $23.50 \rightarrow 25.05$ (DiT, $(\mathbf{x}, \mathbf{y})$) and from $21.68 \rightarrow 24.46$ (MAR, $(\mathbf{x}, \mathbf{y})$), while also improving R3. For camera motion, CLaTr-Score improves from $30.75 \rightarrow 32.81$ (DiT, $(\mathbf{x}, \mathbf{y})$) and from $42.84 \rightarrow 45.96$ (MAR, $(\mathbf{x}, \mathbf{y})$), along with gains in F1. Although modality generation quality metrics ($FD_{TMR}$ and $FD_{CLaTr}$) slightly increase, these changes are modest and do not outweigh the substantial gains in alignment and framing consistency. Overall, Aux achieves a favorable trade-off, improving multimodal coherence and per-modality alignment while maintaining strong generation quality across architectures.

Moreover, we compare our method with baselines for DiT and MAR in Figures 4 and 5, showing the trade-off between framing quality ($FD_{framing}$) and modality-text alignment (TMR for human, CLaTr for camera) across different textual guidance values ($w_c$ in Equation 8). The optimal point lies in the bottom-right corner of each plot (low $FD_{framing}$, high modality scores). Across both architectures and both modalities, our auxiliary sampling achieves the best performance, improving both framing quality and textual alignment, showing its effectiveness and generality.

**Qualitative results.** Figures 6 and 7 show qualitative results with ours sampling for DiT and MAR, respectively. In both cases, the generated human motion is precise, for example, for DiT, the person raises the *right* arm as specified. The camera trajectories are also coherent with the prompt, and accurately following the human motion while maintaining correct on-screen framing, with the subject's head consistently in view. These results highlight that Aux produces humans and cameras that are well aligned with the input prompts, achieving precise motion and coherent framing across different architectures. Further examples are provided on our project page and in the Appendix E.2.

## 5.3 ABLATION STUDY

Table 5: **Auxiliary guidance ablation on the mixed subset.** We vary the auxiliary guidance weight $w_z$ to evaluate its effect on the framing, camera and human metrics. Results are reported for DiT (Peebles & Xie, 2023) and MAR (Li et al., 2024). Superscript $\pm$ denotes the 95% confidence interval over 10 samplings.

| $w_z$ | Framing | | Human | | | | Camera | | | |
|---|---|---|---|---|---|---|---|---|---|---|
| | $FD_{framing}$ ↓ | Out-rate ↓ | $FD_{TMR}$ ↓ | TMR-Score ↑ | R3 ↑ | Coverage ↑ | $FD_{CLaTr}$ ↓ | CLaTr-Score ↑ | F1 ↑ | Coverage ↑ |
| DiT | | | | | | | | | | |
| 0.00 | $4.90^{\pm0.05}$ | $25.98^{\pm0.24}$ | $372.61^{\pm0.90}$ | $23.50^{\pm0.07}$ | $3.67^{\pm0.08}$ | $10.72^{\pm0.15}$ | $87.07^{\pm0.87}$ | $30.75^{\pm0.17}$ | $34.28^{\pm0.27}$ | $51.62^{\pm0.40}$ |
| 0.25 | $3.37^{\pm0.02}$ | $16.76^{\pm0.19}$ | $431.54^{\pm1.15}$ | $25.05^{\pm0.07}$ | $3.89^{\pm0.14}$ | $8.91^{\pm0.13}$ | $80.08^{\pm0.76}$ | $32.81^{\pm0.19}$ | $36.06^{\pm0.25}$ | $48.68^{\pm0.20}$ |
| 0.50 | $3.09^{\pm0.02}$ | $11.99^{\pm0.16}$ | $493.53^{\pm1.64}$ | $25.30^{\pm0.07}$ | $7.23^{\pm0.10}$ | $7.09^{\pm0.17}$ | $90.06^{\pm0.58}$ | $32.45^{\pm0.13}$ | $35.98^{\pm0.09}$ | $44.98^{\pm0.31}$ |
| 0.75 | $3.37^{\pm0.02}$ | $9.66^{\pm0.12}$ | $548.60^{\pm1.62}$ | $24.99^{\pm0.07}$ | $7.08^{\pm0.09}$ | $5.63^{\pm0.15}$ | $123.16^{\pm0.75}$ | $29.58^{\pm0.12}$ | $30.88^{\pm0.17}$ | $38.88^{\pm0.34}$ |
| MAR | | | | | | | | | | |
| 0.00 | $8.51^{\pm0.07}$ | $40.75^{\pm0.28}$ | $275.30^{\pm0.55}$ | $21.68^{\pm0.06}$ | $10.60^{\pm0.19}$ | $17.10^{\pm0.28}$ | $117.77^{\pm0.63}$ | $42.84^{\pm0.14}$ | $42.69^{\pm0.23}$ | $54.89^{\pm0.37}$ |
| 0.50 | $6.42^{\pm0.04}$ | $33.65^{\pm0.23}$ | $301.39^{\pm0.25}$ | $24.46^{\pm0.07}$ | $11.28^{\pm0.09}$ | $14.14^{\pm0.14}$ | $108.74^{\pm0.46}$ | $45.96^{\pm0.14}$ | $45.39^{\pm0.22}$ | $53.67^{\pm0.38}$ |
| 1.00 | $5.93^{\pm0.02}$ | $32.09^{\pm0.17}$ | $326.29^{\pm0.31}$ | $25.42^{\pm0.07}$ | $11.35^{\pm0.14}$ | $12.57^{\pm0.12}$ | $144.02^{\pm0.60}$ | $44.05^{\pm0.16}$ | $40.19^{\pm0.15}$ | $47.26^{\pm0.34}$ |
| 1.50 | $6.04^{\pm0.02}$ | $32.77^{\pm0.15}$ | $346.14^{\pm0.42}$ | $25.65^{\pm0.05}$ | $11.35^{\pm0.20}$ | $11.63^{\pm0.19}$ | $193.14^{\pm0.46}$ | $40.61^{\pm0.11}$ | $36.64^{\pm0.17}$ | $38.21^{\pm0.41}$ |

To assess controllability and effectiveness of auxiliary sampling, we ablate in Table 5 the auxiliary guidance weight $w_z$ (Eq (8)) on both DiT and MAR. We see that a **(1) moderate guidance weight improves framing and text–modality alignment**. On DiT, increasing $w_z$ from 0.00 to 0.25 reduces $FD_{framing}$ $4.90 \rightarrow 3.37$ and Out-rate $25.98 \rightarrow 16.76$; on MAR, $w_z = 0.50$ lowers them $8.51 \rightarrow 6.42$ and $40.75 \rightarrow 33.65$. **(2) Pushing $w_z$ further keeps improving framing but degrades fidelity**: $FD_{TMR}$ and $FD_{CLaTr}$ rise (DiT $431.54 \rightarrow 493.53$, MAR $301.39 \rightarrow 326.29$). **(3) At high weights, it becomes**

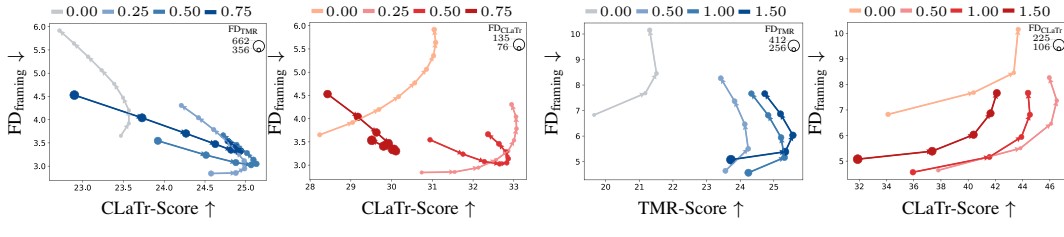

(a) FD$_{\text{framing}}$-TMR-Score  (b) FD$_{\text{framing}}$-CLaTr-Score

(a) FD$_{\text{framing}}$-TMR-Score  (b) FD$_{\text{framing}}$-CLaTr-Score

Figure 8: $w_z$ **ablation in DiT on the mixed set.** Framing quality and modality-text alignment for $c$ guidance ranges from 4 to 12. The optimal region is at the bottom-right (low framing FD, high alignment).

Figure 9: $w_z$ **ablation in MAR on the mixed set.** Framing quality and modality-text alignment for $c$ guidance ranges from 1 to 5. The optimal region is at the bottom-right (low framing FD, high alignment).

**unstable** ($w_z$=0.75 DiT, 1.50 MAR), with FD$_{\text{TMR}}$ spiking to $548.60$ and FD$_{\text{CLaTr}}$ to $193.14$.
We then illustrate Figures 8 and 9 for the trade-off between framing quality (FD$_{\text{framing}}$, lower is better) and text–modality alignment (TMR, CLaTr; higher is better) as the Aux guidance weight $w_z$ varies. The optimum lies near the bottom-right of each plot. Across both architectures, we see: (1) introducing guidance yields a large gain: $w_z$:$0 \rightarrow 0.25$ (DiT) and $0.50$ (MAR) shift points toward the bottom-right; (2) further increases, $0.50$ (DiT), $1.0$ (MAR), continue to improve framing but begin to reduce fidelity, reflected by larger markers (higher Fréchet distances); and (3) at very high weights, $0.75$ (DiT), $1.50$ (MAR), performance degrades on both axes.

**Summary of findings.** From our experiments and ablations, we find that: (1) our proposed sampling consistently improves human–camera coherence (better framing, fewer empty frames) while preserving strong per-modality performance; and (2) the gains generalise across architectures, though absolute performance depends on tuning the guidance weight (as with CFG).

## 6  CONCLUSION

In this paper, we presented a unified framework for joint generation of human motion and camera trajectories, enforcing multimodal coherence via an auxiliary modality: the on-screen framing. Extensive evaluations on the proposed PulpMotion dataset demonstrate the generality and effectiveness of our auxiliary sampling. Our approach currently supports character-level framing but does not yet provide fine-grained control over specific body parts or localized regions, which we identify as an important direction for future work. More broadly, our auxiliary sampling approach is general and could be extended to other modalities and application domains.

ACKNOWLEDGEMENTS

This work was supported by ANR/France 2030 program (*ANR-22-CE23-0007*, *ANR-22-CE39-0016*, *ANR-23-IACL-0005*), Hi!Paris grant, fellowship and chair, DATAIA Convergence Institute as part of the "Programme d'Investissement d'Avenir" (ANR-17-CONV-0003) operated by Ecole Polytechnique, IP Paris, Inria Action Exploratoire PREMEDIT (Precision Medicine using Topology), and the CIEDS.
This project was granted access to the IDRIS High-Performance Computing (HPC) resources under the allocation 2024-AD011013951R2 made by GENCI.
We sincerely thank Yuanzhi Zhu and Nicolas Dufour, for their insightful discussions that contributed to this work and their meticulous proofreading.

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

## A  USE OF LARGE LANGUAGE MODELS

We used large language models solely for text polishing and grammar correction during manuscript preparation. No LLMs were involved in the conception or design of the method, experiments, or analysis. All technical content, results, and conclusions have been independently and carefully verified and validated by the authors.

## B  DETAILED RELATED WORK

**Detailed Human motion generation Related Work.**   Inspired by the success of denoising diffusion models in image generation (Ho et al., 2020; Rombach et al., 2022), several pioneering works (Tevet et al., 2023; Kim et al., 2023; Zhang et al., 2024) adapt diffusion processes to human motion generation. These models are then followed by extensions that leverage pre-trained latent spaces for efficiency, apply consistency distillation for faster sampling, improve caption–motion alignment, and exploit external databases for higher-quality motion (Chen et al., 2023; Dai et al., 2024; Andreou et al., 2025; Zhang et al., 2023b).

While diffusion-based approaches typically represent motion data as raw joint positions and orientations, or continuous latent vectors, another line of work adopts Vector Quantization (VQ) with discrete motion tokens. TM2T (Guo et al., 2022b) first introduce VQ into text-to-motion generation, followed by T2M-GPT (Zhang et al., 2023a), which employed a GPT-style autoregressive model (Brown et al., 2020). More recently, MMM (Pinyoanuntapong et al., 2024), MoMask (Guo et al., 2024), and BiPO (Hong et al., 2024) propose to apply bidirectional attention-based masked generation, inspired by MaskGIT (Chang et al., 2022).

Recently, the masked autoregressive architecture (MAR) (Li et al., 2024) has been proposed to combine the strengths of autoregressive and diffusion models. It has drawn significant attention in the human motion community (Li et al., 2024; Meng et al., 2024; Xiao et al., 2025), as it leverages an autoregressive transformer to handle temporal dynamics while retaining the high generation quality of diffusion models, enabling new state-of-the-art performances.

Nevertheless, most motion generation methods treat motion as an isolated modality, which oversimplifies real-world scenarios where humans continuously interact with their surroundings. Consequently, recent research has begun modeling human interactions with objects (Xu et al., 2023a; Peng et al., 2025b; Geng et al., 2025), other humans (Liang et al., 2024; Fan et al., 2024; Shan et al., 2024), and scenes (Wang et al., 2024b; Cen et al., 2024; Jiang et al., 2024b). However, while recent studies have considered human–camera interaction in motion estimation (Patel & Black, 2025; Ye et al., 2023; Wang et al., 2024a; Kocabas et al., 2024; Sun et al., 2023), motion generation remains largely unexplored, with existing efforts treating camera parameters merely as constraints or conditioning signals rather than modeling their joint distribution with motion.

**Detailed Generative Camera Trajectory generation Related Work.** Over the past two decades, camera control and generation have progressed from handcrafted, rule-based geometric design (Blinn, 1988; Lino & Christie, 2015; Drucker et al., 1992) to deep learning methods that exploit the descriptive and fitting capacity of neural networks: approaches that either learn cinematic rules from example-based references (Jiang et al., 2020; 2021) or leverage the differentiability of deep models to optimize camera trajectories in real-data-supported 3D environments (Wang et al., 2023; Jiang et al., 2024c; Chen et al., 2024).

However, these example-based methods often rely on carefully curated reference videos, and in some cases even synthetic annotation pairs, either to train discriminative models or to optimize trajectories. To mitigate this dependency, other works explore reinforcement learning (RL). In drone cinematography (Huang et al., 2019; Bonatti et al., 2020), RL is guided by human pose and optical flow, while in indoor environments, Xie et al. (2023) propose to use an aesthetic model as the reward function. Though effective within specific environments, both example-based and RL-based methods often collapse into limited trajectory styles and require environment-specific training, resulting in poor generalization.

Yet, with the rapid progress of image and video generative models (Polyak et al., 2024; Peng et al., 2025a; Wang et al., 2025a), a notable direction is to bypass explicit 3D representations and instead treat the model as a universal renderer. This has enabled direct camera-controlled video generation (Wang et al., 2024c; He et al., 2024; Xu et al., 2024; Bahmani et al., 2024; Zheng et al., 2024; Cheong et al., 2024; Wang et al., 2025b). While showing great potential, this line of work faces several limitations: (1) it overlooks scene semantics (e.g., character performance); (2) it still relies on manually designed, complex camera trajectories, which remain challenging for users; and (3) given the relatively low quality of current video generators, the outputs are hard to use directly in production, while the end-to-end nature of these models prevents artists from accessing intermediate assets (e.g., meshes, trajectories, lighting conditions).

To achieve geometric controllability and provide intermediate assets without requiring expert-designed trajectories, Jiang et al. (2024a) introduced the first diffusion-based approach for camera generation. Although limited to synthetic data, their key idea of deriving camera behavior from semantic prompts opens a new direction. Subsequently, Courant et al. (2024) proposed E.T., a large-scale dataset of realistic camera trajectories with human motion from real films, together with evaluation metrics and novel architectural designs. DanceCamAnimator Wang et al. (2024e) and DanceCamera3D Wang et al. (2024d) also focus on dance-specific camera control conditioned on music. More recently, GenDoP (Zhang et al., 2025a) constructed an object-wise, interaction-centric

dataset and employed an autoregressive model to generate trajectories from textual descriptions and visual inputs.

Similarly to human motion generation, most camera generation works condition on human motion but rely solely on global trajectories, which restricts the interaction between camera and subject and overlooks the intrinsic joint distribution problem. In this work, we aim to bridge this gap by addind human motion into the camera trajectory generation pipeline, modelling the symbiosis between how and what to film.

## C  DETAILED METHOD

### C.1  BACKGROUND

#### C.1.1  MOORE-PENROSE PSEUDO-INVERSE AND INDUCED PROJECTIONS

We provide the background defining the Moore-Penrose pseudo-inverse of an $m \times n$ matrix $W$. Let $k := \min\{n, m\}$.

Consider a singular value decomposition of $W$, given by $W = UDV^\top$, where:

1. $U$ is an $m \times m$ orthogonal matrix, i.e., $U^\top U = \boldsymbol{I}_m$,
2. $V$ is an $n \times n$ orthogonal matrix, i.e., $V^\top V = \boldsymbol{I}_n$,
3. $D = \mathrm{diag}(d_1, \ldots, d_k)$ is a $k \times k$ diagonal matrix with non-negative, non-increasing diagonal entries.

The Moore-Penrose pseudo-inverse $W^\dagger$ is then given by:

$$W^\dagger = VD^\dagger U^\top,$$

where $D^\dagger$ is the $k \times k$ diagonal matrix with entries $D_{i,i}^\dagger = d_i^{-1}$ if $d_i > 0$, and 0 otherwise.

Note that if the $k \times k$ matrix $W^\top W$ is invertible, then $D^\dagger = D^{-1}$ and hence $W^\dagger = W^\top (W^\top W)^{-1}$.

Define $P := D^\dagger D$, a diagonal matrix with entries:

$$P_{i,i} = \begin{cases} 1 & \text{if } d_i \neq 0, \\ 0 & \text{otherwise.} \end{cases}$$

We end up with the following properties:

1. *Projection:* $W^\dagger W = VPV^\top$, and $(W^\dagger W)^2 = W^\dagger W$.
2. *Symmetry:* $(W^\dagger W)^\top = W^\dagger W$.
3. *Projection image:* $\ker W = \mathrm{im}(\boldsymbol{I}_n - W^\dagger W)$.

Thus, $P_\perp := \boldsymbol{I}_n - W^\dagger W$ is the orthogonal projection onto $\ker W$, and $P_{/\!/} := W^\dagger W$ is the orthogonal projection onto the orthogonal of $\ker W$: the coimage of $W$.

#### C.1.2  STATISTICS

We start by a well-known special case of the main theorem from Cochran (1934), that we apply to projection matrices.

**Theorem C.1** (Cochran). *Let $X \sim \mathcal{N}(\mu, \sigma^2 \boldsymbol{I}_n)$ be an isotropic Gaussian random vector, and let $F \subseteq \mathbb{R}^n$ be a linear subspace. If $P_F$ and $P_{F\perp}$ are the respective orthogonal projections onto $F$ and its orthogonal complement $F^\perp$, then:*

1. *$P_F X \sim \mathcal{N}(P_F \mu, \sigma^2 P_F)$ and $P_{F\perp} X \sim \mathcal{N}(P_{F\perp} \mu, \sigma^2 P_{F\perp})$ are (possibly degenerate) Gaussian random vectors.*

2. *$P_F X$ and $P_{F\perp} X$ are independent.*

For completeness, we provide a succinct proof below.

*Proof.* Let $P = \begin{bmatrix} P_F \\ P_{F\perp} \end{bmatrix}$. By the properties of Gaussian vectors, $PX$ is a Gaussian vector with covariance matrix:

$$\Sigma = \sigma^2 \begin{bmatrix} P_F \circ P_F^\top & P_F \circ P_{F\top}^\top \\ P_{F\top} \circ P_F^\top & P_{F\top} \circ P_{F\top}^\top \end{bmatrix} = \sigma^2 \begin{bmatrix} P_F & 0 \\ 0 & P_{F\top} \end{bmatrix}, \tag{9}$$

where the last inequality follows from the properties of the orthogonal projections $P_F$ and $P_{F\perp}$, namely: $P_F = P_F^\top$, $P_F^2 = P_F$, and $P_F P_{F\perp} = 0$. Similarly, since the multiplication by $P$ is linear, $\mathbb{E}(PX) = \begin{bmatrix} P_F\mu \\ P_{F\perp}\mu \end{bmatrix}$. Item 1 follows from the block decomposition, and Item 2 follows from Equation (9) and the fact that uncorrelated Gaussian vectors are independent. $\square$

## C.2   METHOD

Let $W^\dagger$ denote the Moore-Penrose pseudo-inverse of $W$ (see Section C.1.1). Using this notation, the matrix $P_\perp := I - W^\dagger W$ (resp. $P_\parallel := W^\dagger W$) can be then identified as the orthogonal projection onto $\ker W$ (resp. $\ker(W)^\perp$). As $u = [\mathbf{x}, \mathbf{y}]^\top \sim \mathcal{N}(\mu, \sigma^2 I_n)$, Cochran's theorem (Theorem C.1) then guarantees that the corresponding projections $\mathbf{u}$:

$$\mathbf{u}_\perp := P_\perp \mathbf{u} \sim \mathcal{N}(P_\perp \mu, \sigma^2 P_\perp) \quad \text{and} \quad \mathbf{u}_\parallel := P_\parallel \mathbf{u} \sim \mathcal{N}(P_\parallel \mu, \sigma^2 P_\parallel), \tag{10}$$

are independent (possibly degenerate) Gaussian vectors.

Observe that $\mathbf{z} = W\mathbf{u} = W\mathbf{u}_\parallel$, so $\mathbf{z}$ is $\mathbf{u}_\parallel$-measurable and thus independent of $\mathbf{u}_\perp$. Moreover, $\mathbf{u}_\parallel = W^\dagger W\mathbf{u} = W^\dagger \mathbf{z}$, which shows that $\mathbf{u}_\parallel$ is a measurable function of $\mathbf{z}$. Therefore,

$$\mathbb{E}[\mathbf{u} \mid \mathbf{z}] = \mathbb{E}\left[\mathbf{u}_\perp + W^\dagger \mathbf{z} \mid \mathbf{z}\right] \stackrel{\mathbf{u}_\perp \text{ indep. } \mathbf{z}}{=} \mathbb{E}(\mathbf{u}_\perp) + \mathbb{E}[W^\dagger \mathbf{z} \mid \mathbf{z}] \stackrel{\mathbf{z}\text{-meas.}}{=} P_\perp \mu + W^\dagger \mathbf{z}. \tag{11}$$

We have the following decomposition of $\mathbf{u}$ into two independent variables: $\mathbf{u} = P_\perp \mathbf{u} + P_\parallel \mathbf{u} = P_\perp \mathbf{u} + W^\dagger \mathbf{z}$. This induces a decomposition of the density $p_\mathbf{u}$ of $\mathbf{u}$ into two density functions[2] $p_{\mathbf{u}_\perp}$ and $p_{\mathbf{u}_\parallel} = p_{W^\dagger \mathbf{z}}$ of $\mathbf{u}_\perp$ and $\mathbf{u}_\parallel = W^\dagger \mathbf{z}$ respectively. Given point $u \in \mathbb{R}^n$:

$$p_\mathbf{u}(u) = p_{(\mathbf{u}_\perp, \mathbf{u}_\parallel)}(u_\perp, u_\parallel) \stackrel{\text{indep.}}{=} p_{\mathbf{u}_\perp}(u_\perp) p_{\mathbf{u}_\parallel}(u_\parallel), \tag{12}$$

where the functions $p_{\mathbf{u}_\perp}$ and $p_{\mathbf{u}_\parallel}$ are given by:

$$v \mapsto p_{\mathbf{u}_\perp}(v) = \frac{\mathbb{1}_{\ker W}(v)}{\sqrt{2\pi\sigma^2}^{\dim \ker W}} \exp\left[-\frac{1}{2\sigma^2}(v-\mu)^\top(v-\mu)\right], \text{ and}$$

$$v \mapsto p_{\mathbf{u}_\parallel}(v) = \frac{\mathbb{1}_{\ker(W)^\perp}(v)}{\sqrt{2\pi\sigma^2}^{\dim \ker(W)^\perp}} \exp\left[-\frac{1}{2\sigma^2}(v-\mu)^\top(v-\mu)\right].$$

## C.3   MORE DETAILS ON THE DENSITY DECOMPOSITION

We give here more details on the possible decompositions of the density $p_\mathbf{u}$ of $\mathbf{u}$. The decomposition given in Equation (12) from ensures that we have for a given point $u \in \mathbb{R}^n$:

$$p_\mathbf{u}(u) = p_{\mathbf{u}_\perp}((I - W^\dagger W)u) p_{\mathbf{u}_\parallel}(W^\dagger W u). \tag{13}$$

Following the argumentation of Equation (11), given a point $z \in \mathbb{R}^m$, the conditional distribution $\mathbf{u} \mid \mathbf{z} = z$ of $\mathbf{u}$ conditionally to the event $\{\mathbf{z} = z\}$ is given by $\mathbf{u}_\perp + W^\dagger z$: a translation of $\mathbf{u}_\perp$ by the constant $W^\dagger z$ and thus admits a density $p_{\mathbf{u}|\mathbf{z}=z}$, given by:

$$(u, z) \longmapsto p_{\mathbf{u}|\mathbf{z}=z}(u) = p_{\mathbf{u}_\perp + W^\dagger z}(u) = p_{\mathbf{u}_\perp}(u - W^\dagger z).$$

---

[2]Note that $\mathbf{u}_\perp$ and $\mathbf{u}_\parallel$ do not admit densities w.r.t. the Lebesgue measure on $\mathbb{R}^n$, since they are supported on the non-full-dimensional vector spaces $\ker W$ and $\ker(W)^\perp$ respectively. Nevertheless, they admit densities $p_{\mathbf{u}_\perp}$ and $p_{\mathbf{u}_\parallel}$ w.r.t. the respective Lebesgue measures on these subspaces. See Section C.3 for more details.

This can be directly shown by the change of variable $u = u_\perp + W^\dagger z = u_\perp + W^\dagger W u_\parallel$ in Equation (13), since on one hand, we have $z = Wu$ almost surely, hence

$$p_{\mathbf{u}}(u) = p_{(\mathbf{u}_\perp, \mathbf{u}_\parallel)}(u_\perp, u_\parallel) \overset{\text{indep.}}{=} p_{\mathbf{u}_\perp}(u_\perp)p_{\mathbf{u}_\parallel}(u_\parallel) \overset{\text{shift}}{=} p_{\mathbf{u}_\perp + u_\parallel}(u_\perp + u_\parallel)p_{\mathbf{u}_\parallel}(u_\parallel).$$

Furthermore, on the other hand, we have $\mathbf{u}_\parallel = W^\dagger \mathbf{z}$ and $\mathbf{z} = W\mathbf{u}$, hence $p_{\mathbf{z}}(\cdot) \propto p_{\mathbf{u}_\parallel}(W\cdot)$, and therefore, for any point $u$ and $z = Wu$:

$$p_{\mathbf{u}}(u) \overset{z=Wu}{=} p_{\mathbf{u}_\perp + W^\dagger z}(u)p_{\mathbf{u}_\parallel}(W^\dagger z) \overset{z=Wu}{\propto} p_{\mathbf{u}_\perp + W^\dagger z}(u)p_{\mathbf{z}}(z).$$

This equality being true for (almost) every $u$ and $z = Wu$, we conclude that $p_{\mathbf{u}|\mathbf{z}=z} \propto p_{\mathbf{u}_\perp + W^\dagger z}$ almost everywhere. This property can be shown more directly by invoking the fact that the $\sigma$-algebra generated by $\mathbf{z}$ and the one of $\mathbf{u}_\parallel$ are the same.

### C.4 AUXILIARY SAMPLING DERIVATION

In this section, we detail the derivation from Equation (14) to Equation (8). Starting from Equation (4) and applying the decomposition in Equation (6), we have:

$$\begin{aligned}
\nabla_{\mathbf{x}_t, \mathbf{y}_t} \log \tilde{p}(\mathbf{x}_t, \mathbf{y}_t | \mathbf{c}) = {} & \nabla_{\mathbf{x}_t, \mathbf{y}_t} \log p(\mathbf{u}_\perp) + (1 + w_z)\nabla_{\mathbf{x}_t, \mathbf{y}_t} \log p(\mathbf{u}_\parallel) \\
& + w_c(\nabla_{\mathbf{x}_t, \mathbf{y}_t} \log p(\mathbf{x}_t, \mathbf{y}_t | \mathbf{c}) - \nabla_{\mathbf{x}_t, \mathbf{y}_t} \log p(\mathbf{x}_t, \mathbf{y}_t)).
\end{aligned} \tag{14}$$

Therefore, since $\begin{bmatrix} \mathbf{x} \\ \mathbf{y} \end{bmatrix} \sim \mathcal{N}(\boldsymbol{\mu}, \sigma^2 \boldsymbol{I})$ we obtain:

$$\begin{aligned}
\nabla_{\mathbf{x}_t, \mathbf{y}_t} \log p(\mathbf{x}_t, \mathbf{y}_t | \mathbf{c}) = {} & -\frac{1}{\sigma^2}(\boldsymbol{I} - \boldsymbol{P}_\parallel)\left([\mathbf{x}_t, \mathbf{y}_t]^\top - \boldsymbol{\mu}\right) - \frac{1}{\sigma^2}(1 + w_z)\boldsymbol{P}_\parallel\left([\mathbf{x}_t, \mathbf{y}_t]^\top - \boldsymbol{\mu}\right) \\
& - \frac{1}{\sigma^2}w_c\left((\mathbf{x}_c - \boldsymbol{\mu}_c) - ([\mathbf{x}_t, \mathbf{y}_t]^\top - \boldsymbol{\mu})\right) \\
= {} & -\frac{1}{\sigma^2}\left([\mathbf{x}_t, \mathbf{y}_t]^\top - \boldsymbol{\mu}\right) \\
& - \frac{1}{\sigma^2}w_z \boldsymbol{P}_\parallel\left([\mathbf{x}_t, \mathbf{y}_t]^\top - \boldsymbol{\mu}\right) \\
& - \frac{1}{\sigma^2}w_c\left(([\mathbf{x}_t^c, \mathbf{y}_t^c]^\top - \boldsymbol{\mu}_c) - ([\mathbf{x}_t, \mathbf{y}_t]^\top - \boldsymbol{\mu})\right).
\end{aligned} \tag{15}$$

Finally, recalling that $\boldsymbol{\varepsilon} = -\frac{1}{\sigma}(\mathbf{x} - \boldsymbol{\mu})$, we can express the sampling equation with noise prediction as:

$$\begin{aligned}
\boldsymbol{\varepsilon}(\mathbf{x}_t, \mathbf{y}_t, \mathbf{c}, t) = {} & \boldsymbol{\varepsilon}(\mathbf{x}_t, \mathbf{y}_t, \emptyset, t) \\
& + w_z \boldsymbol{P}_\parallel \, \boldsymbol{\varepsilon}(\mathbf{x}_t, \mathbf{y}_t, \emptyset, t) \\
& + w_c(\boldsymbol{\varepsilon}(\mathbf{x}_t, \mathbf{y}_t, \mathbf{c}, t) - \boldsymbol{\varepsilon}(\mathbf{x}_t, \mathbf{y}_t, \emptyset, t)).
\end{aligned} \tag{16}$$

## D DETAILED DATASET

### D.1 DETAILED PIPELINE

In this section, we describe the construction of the PulpMotion dataset, illustrated in Figure 10. We first apply an off-the-shelf camera-human pose estimator (Wang et al., 2024a) to infer both camera and human poses from video clips of the CondensedMovies dataset (Bain et al., 2020). As noted in the main manuscript, a key challenge of video-based pose estimation is handling occluded or unseen body parts, which are often inaccurately predicted.

To address this, we first identify poorly estimated regions by reprojecting visible joints. We then use a vision-language model (VLM) to generate captions describing human motion, providing bounding boxes around the target person to guide the model's focus. We show an example of human motion caption generation in Figure 26 with input prompt and VLM response.

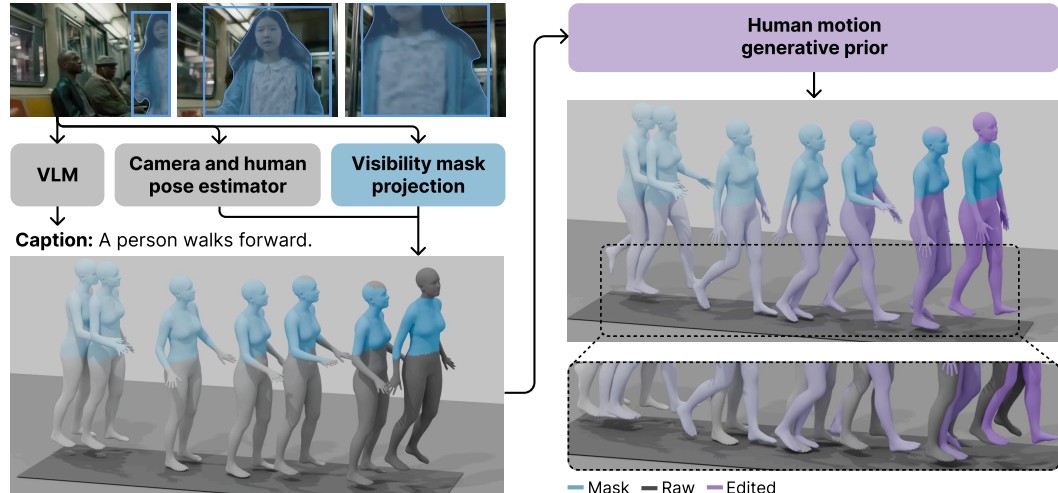

Figure 10: **Dataset refinement pipeline.** Given RGB frames from a video, we first estimate the camera and human pose. We then identify the out-of-screen body parts by reprojection. Finally, we refine the out-of-screen parts using a generative prior.

Table 6: **Comparison of PulpMotion and E.T. (Courant et al., 2024) datasets.** We compare the full (*all*), *pure*, and *mixed* subsets of PulpMotion with the E.T.. We summarize modality coverage, available captions, dataset size (hours, frames, samples), sample length statistics (median, mean, std), and vocabulary size.

| Dataset | Camera | | Human | | #Hours | #Frames | #Samples | Sample lengths (frames) | | | #Vocabulary |
|---|---|---|---|---|---|---|---|---|---|---|---|
| | Traj | Caption | Motion | Caption | | | | Median | Mean | Std | |
| E.T. Courant et al. (2024) | | | | | | | | | | | |
| *all* | | | | | 120 | 11M | 115K | 75 | 93.9 | 73.8 | 1,790 |
| *pure* | ✓ | ✓ | ✓ | ✗ | 20 | 1.8M | 30K | 46 | 59.5 | 49.1 | 941 |
| *mixed* | | | | | 67 | 6M | 65K | 72 | 92.9 | 75.06 | 1,579 |
| PulpMotion (Ours) | | | | | | | | | | | |
| *all* | | | | | 314 | 22M | 193K | 107 | 117.3 | 63.6 | 4,599 |
| *pure* | ✓ | ✓ | ✓ | ✓ | 51 | 3.7M | 41K | 70 | 91.1 | 59.2 | 2,831 |
| *mixed* | | | | | 170 | 12M | 105K | 108 | 116.44 | 60.44 | 4,143 |

Next, in the right part of Figure 10, we refine the occluded regions using a diffusion-based editing method (Lugmayr et al., 2022) with a model pretrained on HumanML3D (Guo et al., 2022a). To avoid artifacts caused by naive editing, we refine the entire sub-kinematic chain of each occluded joint rather than modifying joints in isolation. Since visible parts remain largely unchanged, projection consistency between the reconstructed body and RGB frames is preserved.

## D.2 DETAILED STATISTICS

Table 6 compares our PulpMotion dataset with E.T. (Courant et al., 2024) across several dimensions.

Overall, PulpMotion significantly increases the dataset size, containing 314 hours and 22M frames compared to E.T.'s 120 hours and 11M frames. Our dataset also provides longer samples (median 107 frames vs. 75). The "pure" and "mixed" subsets follow the same trends, demonstrating consistent improvements.

Thanks to our refinement pipeline, as shown in the main manuscript, PulpMotion ensures higher-quality human motions. Additionally, PulpMotion includes HumanML3D-style human motion captions, which are not available in E.T.

# E   DETAILED EXPERIMENTS

## E.1   AUTOENCODER

### E.1.1   DETAILED EXPERIMENTAL SETUP

**Implementation details.**   We adopt the ResNet-based autoencoder from MARDM (Meng et al., 2024) with ReLU activations. The joint encoder and three modality-specific decoders have temporal down-/up-sampling by a factor of 4, and each consist of two 1D-ResNet blocks. Latent dimensions are set to $64$ for the camera, $128$ for the human, and $64$ for the projection. The model is trained for 325 epochs on the full Pulp Motion dataset with 64-frame samples (and evaluated on 300-frame samples), using AdamW with a learning rate of $1.9 \times 10^{-4}$, a batch size of $128$, on a single A100 GPU. A linear warmup of 1K steps is applied, followed by a decay of 0.1 after 4K steps.

### E.1.2   DETAILED PERFORMANCES

Table 7: **Reconstruction evaluation of autoencoder.** We report reconstruction metrics for *pure* and *mixed* subsets. Metrics span projection accuracy (MPJProjE, $FD_{framing}$), human pose quality (MPJPE, FDTMR, TMR-Score), and camera alignment (APE, $FD_{CLaTr}$, CLaTr-Score).

| Methods | Framing | | Human | | | Camera | | |
|---|---|---|---|---|---|---|---|---|
| | MPJProjPE ↓ | $FD_{framing}$ ↓ | MPJPE ↓ | $FD_{TMR}$ ↓ | TMR-Score ↑ | APE ↓ | $FD_{CLaTr}$ ↓ | CLaTr-Score ↑ |
| *pure* | | | | | | | | |
| Ground truth | 0.00 | 0.00 | 0.00 | 0.00 | 16.47 | 0.00 | 0.00 | 70.25 |
| AE | 0.09 | 0.14 | 3.26 | 105.57 | 15.93 | 0.15 | 19.26 | 60.45 |
| *mixed* | | | | | | | | |
| Ground truth | 0.00 | 0.00 | 0.00 | 0.00 | 17.72 | 0.00 | 0.00 | 68.88 |
| AE | 0.08 | 0.23 | 5.63 | 124.78 | 18.16 | 0.18 | 15.64 | 57.98 |

We evaluate the reconstruction quality of the autoencoder introduced in Section 3.1 using modality-specific errors: mean per-joint projected error (MPJProjPE) and mean per-joint error (MPJPE) for human motion, and absolute pose error (APE) for the camera. Additionally, we compute reconstruction Fréchet distances and modality–text alignment metrics from Section 5.1.

As shown in Table 7, the autoencoder achieves low MPJProjPE (0.08–0.09), indicating reliable 2D frame reconstruction across both subsets. MPJPE reveals discrepancies in 3D pose recovery, particularly in the mixed setting (5.63 vs. 3.26). APE remains low ($\leq 0.18$) but shows slight degradation in the mixed case, consistent with the observed drop in CLaTr-Score.

## E.2   GENERATION

### E.2.1   DETAILED EXPERIMENTAL SETUP

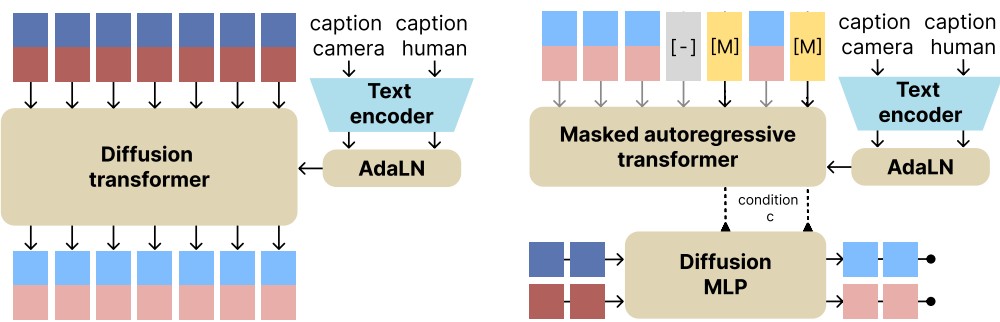

Figure 11: **Overview of the DiT architecture.**   Figure 12: **Overview of the MAR architecture.**

We illustrate in Figures 12 and 11 both DiT Peebles & Xie (2023) and MAR Li et al. (2024) architectures used in this work.

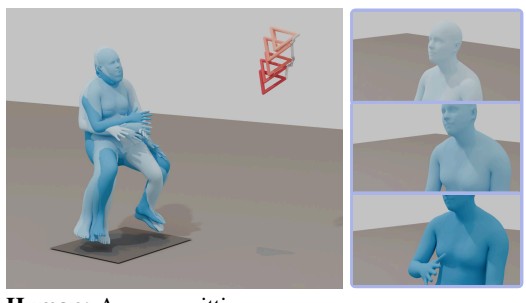 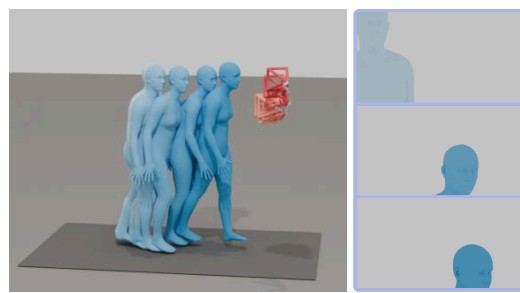

**Human:** A person sitting.
**Camera:** The camera performs a boom bottom.

Figure 13: **Example with DiT on the mixed set.**

**Human:** A person walks while turning head to right.
**Camera:** The camera booms up.

Figure 14: **Example with MAR on the mixed set.**

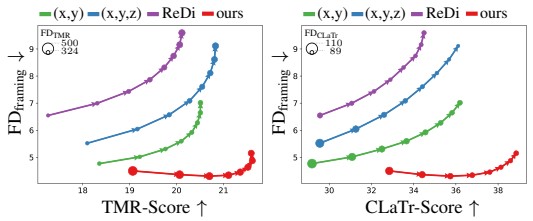 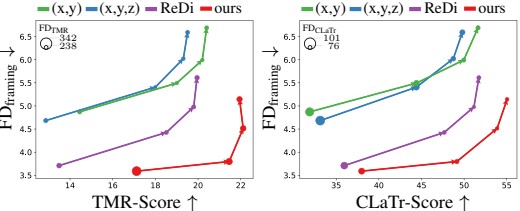

(a) FD_framing-TMR-Score  (b) FD_framing-CLaTr-Score

(a) FD_framing-TMR-Score  (b) FD_framing-CLaTr-Score

Figure 15: **State-of-the-art comparison in DiT on the pure subset.** Trade-off between framing quality and modality-text alignment for textual guidance ranges from 5 to 12. The optimal region is at the bottom-right (low framing error, high alignment).

Figure 16: **State-of-the-art comparison in MAR on the pure subset.** Trade-off between framing quality and modality-text alignment for textual guidance values ranges from 2 to 5. The optimal region is at the bottom-right (low framing error, high alignment).

**Implementation details.** We evaluate two architectures: a DiT-based model with in-context conditioning (Courant et al., 2024) and a MAR-based model with AdaLN conditioning (Meng et al., 2024). To ensure fairness, both are scaled to $\sim 28.3$M parameters. The DiT model has 8 layers with a hidden dimension 532 and 14 attention heads. The MAR model uses a single-layer autoregressive transformer (hidden dimension 512, 8 heads) and a diffusion head with 3 MLP layers of width 1024. Both models are trained for 93k steps on the *pure* subset and 330K steps on the *mixed* subset on the pure and mixed subsets of Pulp Motion with 300-frame samples, using AdamW with a learning rate of $3 \times 10^{-4}$, a batch size of 128, on a single A100 GPU. A linear warmup of 2K steps is applied, followed by decay by 0.1 after 50K steps. For inference, we perform 50 DDPM sampling steps.

### E.2.2 MORE QUALITATIVE RESULTS ON MIXED DATASET

We show in Figure 13 and Figure 14 additional qualitative results on the *mixed* subset. We also provide additional qualitative videos on our project page including

- **Generation examples**, comparing our method to baselines on both DiT and MAR architectures. We observe that our approach achieves more stable and consistent framing, reliably keeping the human on screen, whereas baselines either ignore the subject entirely or fail to maintain framing over the full sequence.

- **Dataset pipeline**, illustrating the extraction of camera, human, and textual information as well as the refinement step, which noticeably improves motion naturalness (e.g., converting sliding artifacts into realistic walking).

- **Dataset examples**, highlighting the diversity of camera trajectories, human motions, and textual captions.

Table 8: **State-of-the-art comparison on the pure subset.** We compare five baselines: human-conditioned camera generation $(\mathbf{x})$+DIRECTOR Courant et al. (2024), independent modality generation $(\mathbf{x})(\mathbf{y})$, dual-modality generation $(\mathbf{x}, \mathbf{y})$, triplet-modality generation $(\mathbf{x}, \mathbf{y}, \mathbf{z})$, and ReDi Kouzelis et al. (2025), along with our auxiliary sampling (Aux) method. Results are reported for DiT (Peebles & Xie, 2023) and MAR (Li et al., 2024). Superscript $\pm$ denotes the 95% confidence interval over 10 samplings.

| Methods | Framing | | Human | | | | Camera | | | |
|---|---|---|---|---|---|---|---|---|---|---|
| | $FD_{framing}$ ↓ | Out-rate ↓ | $FD_{TMR}$ ↓ | TMR-Score ↑ | R3 ↑ | Coverage ↑ | $FD_{CLaTr}$ ↓ | CLaTr-Score ↑ | F1 ↑ | Coverage ↑ |
| Ground-truth | 0.00 | 0.71 | 0.00 | 16.47 | 19.79 | 1.00 | 0.00 | 70.25 | 94.52 | 1.0 |
| Auto-encoder | 0.14 | 3.46 | 105.57 | 15.93 | 20.21 | 89.00 | 19.26 | 60.45 | 77.51 | 78.96 |
| **DiT** | | | | | | | | | | |
| $(\mathbf{x})$+DIRECTOR | $23.58^{\pm0.07}$ | $67.94^{\pm0.24}$ | - | - | - | - | $127.19^{\pm0.89}$ | $34.47^{\pm0.32}$ | $42.97^{\pm0.57}$ | $40.58^{\pm0.23}$ |
| $(\mathbf{x})(\mathbf{y})$ | $10.24^{\pm0.08}$ | $41.70^{\pm0.40}$ | $384.31^{\pm0.62}$ | $23.72^{\pm0.10}$ | $20.63^{\pm0.35}$ | $10.46^{\pm0.18}$ | $86.06^{\pm0.38}$ | $57.74^{\pm0.29}$ | $75.53^{\pm0.23}$ | $31.83^{\pm0.21}$ |
| $(\mathbf{x})(\mathbf{y})$+Aux (ours) | $7.88^{\pm0.07}$ | $36.03^{\pm0.36}$ | $443.65^{\pm0.89}$ | $24.67^{\pm0.09}$ | $21.41^{\pm0.35}$ | $8.63^{\pm0.19}$ | $77.78^{\pm0.57}$ | $62.75^{\pm0.31}$ | $83.00^{\pm0.35}$ | $29.53^{\pm0.28}$ |
| $(\mathbf{x}, \mathbf{y})$ | $6.78^{\pm0.06}$ | $36.25^{\pm0.36}$ | $372.75^{\pm0.94}$ | $20.74^{\pm0.10}$ | $18.16^{\pm0.25}$ | $12.73^{\pm0.31}$ | $93.37^{\pm0.78}$ | $35.99^{\pm0.20}$ | $48.82^{\pm0.39}$ | $44.56^{\pm0.33}$ |
| $(\mathbf{x}, \mathbf{y}, \mathbf{z})$ | $5.56^{\pm0.06}$ | $29.81^{\pm0.27}$ | $334.29^{\pm1.10}$ | $18.04^{\pm0.19}$ | $15.52^{\pm0.22}$ | $17.46^{\pm0.25}$ | $108.05^{\pm1.21}$ | $28.62^{\pm0.35}$ | $41.91^{\pm0.41}$ | $45.83^{\pm0.37}$ |
| $(\mathbf{x}, \mathbf{y}, \mathbf{z})$+Aux (ours) | $4.66^{\pm0.04}$ | $23.61^{\pm0.36}$ | $438.38^{\pm1.44}$ | $19.47^{\pm0.10}$ | $4.85^{\pm0.19}$ | $14.78^{\pm0.21}$ | $83.65^{\pm1.29}$ | $30.80^{\pm0.31}$ | $41.30^{\pm0.59}$ | $47.06^{\pm0.34}$ |
| ReDi | $6.53^{\pm0.05}$ | $33.34^{\pm0.27}$ | $323.53^{\pm0.74}$ | $17.13^{\pm0.19}$ | $15.21^{\pm0.30}$ | $17.81^{\pm0.19}$ | $99.60^{\pm0.92}$ | $28.65^{\pm0.36}$ | $40.49^{\pm0.41}$ | $48.60^{\pm0.33}$ |
| $(\mathbf{x}, \mathbf{y})$+Aux (ours) | $5.03^{\pm0.03}$ | $24.92^{\pm0.28}$ | $424.81^{\pm1.07}$ | $21.80^{\pm0.12}$ | $18.32^{\pm0.15}$ | $11.69^{\pm0.19}$ | $91.36^{\pm0.81}$ | $38.42^{\pm0.31}$ | $51.61^{\pm0.47}$ | $40.94^{\pm0.19}$ |
| **MAR** | | | | | | | | | | |
| $(\mathbf{x})(\mathbf{y})$ | $11.22^{\pm0.04}$ | $45.39^{\pm0.48}$ | $261.20^{\pm0.78}$ | $21.66^{\pm0.10}$ | $27.59^{\pm0.40}$ | $18.89^{\pm0.23}$ | $120.50^{\pm0.81}$ | $57.18^{\pm0.23}$ | $68.09^{\pm0.24}$ | $38.97^{\pm0.54}$ |
| $(\mathbf{x})(\mathbf{y})$+Aux (ours) | $9.32^{\pm0.07}$ | $41.54^{\pm0.36}$ | $280.36^{\pm0.84}$ | $23.25^{\pm0.06}$ | $28.56^{\pm0.27}$ | $15.79^{\pm0.17}$ | $108.43^{\pm0.66}$ | $60.74^{\pm0.12}$ | $71.08^{\pm0.48}$ | $34.60^{\pm0.32}$ |
| $(\mathbf{x}, \mathbf{y})$ | $6.55^{\pm0.10}$ | $30.19^{\pm0.34}$ | $251.94^{\pm1.46}$ | $20.16^{\pm0.13}$ | $25.48^{\pm0.29}$ | $28.25^{\pm0.43}$ | $108.28^{\pm1.83}$ | $52.17^{\pm0.32}$ | $67.31^{\pm0.49}$ | $55.48^{\pm0.52}$ |
| $(\mathbf{x}, \mathbf{y}, \mathbf{z})$ | $6.10^{\pm0.11}$ | $30.11^{\pm0.32}$ | $242.81^{\pm0.91}$ | $19.23^{\pm0.10}$ | $25.17^{\pm0.46}$ | $30.33^{\pm0.48}$ | $116.75^{\pm1.04}$ | $49.52^{\pm0.23}$ | $63.14^{\pm0.37}$ | $55.81^{\pm0.50}$ |
| $(\mathbf{x}, \mathbf{y}, \mathbf{z})$+Aux (ours) | $4.30^{\pm0.05}$ | $23.48^{\pm0.28}$ | $262.34^{\pm0.61}$ | $21.08^{\pm0.14}$ | $10.30^{\pm0.30}$ | $19.26^{\pm0.34}$ | $108.98^{\pm0.59}$ | $52.36^{\pm0.38}$ | $66.16^{\pm0.47}$ | $49.72^{\pm0.46}$ |
| ReDi | $5.07^{\pm0.10}$ | $25.84^{\pm0.31}$ | $252.58^{\pm0.92}$ | $19.73^{\pm0.11}$ | $25.50^{\pm0.41}$ | $28.34^{\pm0.37}$ | $103.13^{\pm1.14}$ | $51.99^{\pm0.27}$ | $66.95^{\pm0.38}$ | $56.29^{\pm0.59}$ |
| $(\mathbf{x}, \mathbf{y})$+Aux (ours) | $4.90^{\pm0.06}$ | $24.28^{\pm0.31}$ | $281.39^{\pm0.83}$ | $21.90^{\pm0.17}$ | $26.43^{\pm0.40}$ | $17.48^{\pm0.26}$ | $100.66^{\pm0.92}$ | $55.43^{\pm0.27}$ | $69.76^{\pm0.50}$ | $47.87^{\pm0.44}$ |

### E.2.3 EXTRA COMPARISON TO THE STATE OF THE ART ON PURE DATASET

**Quantitative results.** In the main manuscript, we focused on the *mixed* dataset. This section reports additional experiments trained and evaluated on the *pure* subset. Table 8 reports a comparison of our auxiliary sampling (Aux) method against state-of-the-art baselines across both DiT and MAR architectures on the pure subset. We summarize our experimental observations as follows:

**(i) Auxiliary sampling consistently improves framing (multimodal coherence).** Applying Aux leads to systematic improvements in framing quality ($FD_{framing}$) and out-of-frame (Out-rate) rates across all baseline configurations and architectures. For DiT, auxiliary sampling reduces $FD_{framing}$ from $11.24 \rightarrow 7.88$ for $(\mathbf{x})(\mathbf{y})$, from $6.78 \rightarrow 5.03$ when applied to $(\mathbf{x}, \mathbf{y})$. A similar trend holds for MAR, where $FD_{framing}$ decreases from $11.22 \rightarrow 9.32$ for $(\mathbf{x})(\mathbf{y})$ and from $6.55 \rightarrow 4.90$ for $(\mathbf{x}, \mathbf{y})$. Out rates follow the same pattern: for DiT, Aux reduces the out rate from $41,70\% \rightarrow 36.03\%$ $((\mathbf{x})(\mathbf{y}))$ and from $36.25\% \rightarrow 24,92\%$ $((\mathbf{x}, \mathbf{y}))$, while for MAR it decreases from $45.39\% \rightarrow 41,54\%$ and $30.19\% \rightarrow 24.28\%$, respectively. Overall, auxiliary sampling achieves the best $FD_{framing}$ and out rates across both architectures (DiT and MAR): showing that it is architecture-agnostic and consistently enhances framing quality, i.e. multimodal coherence, for all settings (independent and joint).

**(ii) Conditioning on human motion alone is insufficient for strong framing.** We next examine the human-conditioned camera generation baseline $(\mathbf{x})$+DIRECTOR, where human trajectories are first generated using the same backbone as the independent setting $((\mathbf{x})(\mathbf{y}))$, and camera motion is subsequently conditioned on these synthesized humans. While this strategy is weaker than other baselines: for example, under DiT, $(\mathbf{x})$+DIRECTOR yields a $FD_{framing}$ of $23.58$, compared to $7.88$ for $(\mathbf{x})(\mathbf{y})$+Aux and $5.03$ for $(\mathbf{x}, \mathbf{y})$+Aux. Its out rate is also higher with $67.94\%$ against $36.03\%$ and $24.92\%$, respectively. These results indicate that conditioning on human motion alone provides limited contextual information for accurate camera framing. In contrast, joint generation methods, especially when combined with Aux, consistently outperform $(\mathbf{x})$+DIRECTOR and the independent baseline $((\mathbf{x})(\mathbf{y}))$, highlighting the importance of jointly modeling modalities during sampling.

**(iii) Auxiliary sampling strengthens per-modality alignment while preserving quality.** Beyond framing, Aux improves text–modality alignment across both human and camera dimensions. For human motion, auxiliary sampling increases TMR-Score from $20.74 \rightarrow 21.80$ (DiT, $(\mathbf{x}, \mathbf{y})$) and from $20.16 \rightarrow 21.90$ (MAR, $(\mathbf{x}, \mathbf{y})$), while also improving R3. For camera motion, CLaTr-Score improves from $35.99 \rightarrow 38.42$ (DiT, $(\mathbf{x}, \mathbf{y})$) and from $52.17 \rightarrow 55.43$ (MAR, $(\mathbf{x}, \mathbf{y})$), along with gains in F1. Although modality generation quality metrics ($FD_{TMR}$ and $FD_{CLaTr}$) slightly increase, these changes are modest and do not outweigh the substantial gains in alignment and framing consistency. Overall, Aux achieves a favorable trade-off, improving multimodal coherence and per-modality alignment while maintaining strong generation quality across architectures.

Moreover, we compare our method with baselines for DiT and MAR in Figures 15 and 16, showing the trade-off between framing quality ($FD_{framing}$) and modality-text alignment (TMR for human,

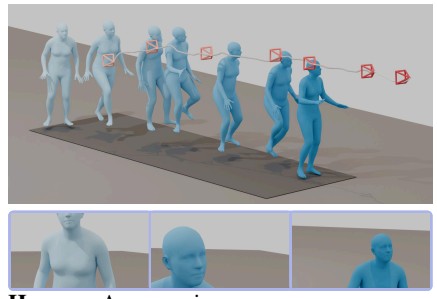 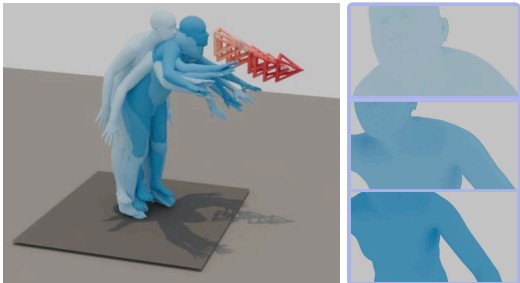

**Human:** A person jumps.
**Camera:** The camera performs a pull out.

**Human:** A person leans forward.
**Camera:** The camera performs a pull out.

Figure 17: **Example with DiT on the pure subset.**  Figure 18: **Example with MAR on the pure subset.**

Table 9: **Auxiliary guidance ablation on the pure subset.** We vary the auxiliary guidance weight $w_z$ to evaluate its effect on the framing, camera and human metrics. Results are reported for DiT (Peebles & Xie, 2023) and MAR (Li et al., 2024). Superscript $\pm$ denotes the 95% confidence interval over 10 samplings.

| $w_z$ | Framing | | Human | | | | Camera | | | |
|---|---|---|---|---|---|---|---|---|---|---|
| | $FD_{framing}$ ↓ | Out-rate ↓ | $FD_{TMR}$ ↓ | TMR-Score ↑ | R3 ↑ | Coverage ↑ | $FD_{CLaTr}$ ↓ | CLaTr-Score ↑ | F1 ↑ | Coverage ↑ |
| DiT | | | | | | | | | | |
| 0.00 | $6.78^{\pm0.06}$ | $36.25^{\pm0.36}$ | $372.75^{\pm0.94}$ | $20.74^{\pm0.10}$ | $18.16^{\pm0.25}$ | $12.73^{\pm0.31}$ | $93.37^{\pm0.78}$ | $35.99^{\pm0.20}$ | $48.82^{\pm0.39}$ | $44.56^{\pm0.33}$ |
| 0.25 | $5.03^{\pm0.03}$ | $24.92^{\pm0.28}$ | $424.81^{\pm1.07}$ | $21.80^{\pm0.12}$ | $18.32^{\pm0.15}$ | $11.69^{\pm0.19}$ | $91.36^{\pm0.81}$ | $38.42^{\pm0.31}$ | $51.61^{\pm0.47}$ | $40.94^{\pm0.19}$ |
| 0.50 | $4.27^{\pm0.03}$ | $17.15^{\pm0.30}$ | $460.87^{\pm1.33}$ | $21.84^{\pm0.11}$ | $18.80^{\pm0.25}$ | $8.76^{\pm0.15}$ | $111.37^{\pm0.74}$ | $37.87^{\pm0.23}$ | $50.35^{\pm0.33}$ | $37.52^{\pm0.27}$ |
| 0.75 | $4.52^{\pm0.03}$ | $13.84^{\pm0.32}$ | $510.14^{\pm1.41}$ | $21.54^{\pm0.12}$ | $18.28^{\pm0.27}$ | $6.99^{\pm0.16}$ | $152.87^{\pm1.01}$ | $34.84^{\pm0.28}$ | $44.22^{\pm0.39}$ | $32.80^{\pm0.29}$ |
| MAR | | | | | | | | | | |
| 0.00 | $6.55^{\pm0.10}$ | $30.19^{\pm0.34}$ | $251.94^{\pm1.46}$ | $20.16^{\pm0.13}$ | $25.48^{\pm0.29}$ | $28.25^{\pm0.43}$ | $108.28^{\pm1.83}$ | $52.17^{\pm0.32}$ | $67.31^{\pm0.49}$ | $55.48^{\pm0.52}$ |
| 0.50 | $4.90^{\pm0.06}$ | $24.28^{\pm0.31}$ | $281.39^{\pm0.83}$ | $21.90^{\pm0.17}$ | $26.43^{\pm0.40}$ | $17.48^{\pm0.26}$ | $100.66^{\pm0.92}$ | $55.43^{\pm0.27}$ | $69.76^{\pm0.50}$ | $47.87^{\pm0.44}$ |
| 1.00 | $4.62^{\pm0.04}$ | $22.85^{\pm0.26}$ | $308.10^{\pm1.03}$ | $22.46^{\pm0.14}$ | $26.43^{\pm0.32}$ | $15.36^{\pm0.16}$ | $134.96^{\pm0.98}$ | $52.74^{\pm0.28}$ | $61.26^{\pm0.30}$ | $41.28^{\pm0.38}$ |
| 1.50 | $4.75^{\pm0.03}$ | $23.19^{\pm0.36}$ | $330.03^{\pm0.97}$ | $22.63^{\pm0.12}$ | $26.44^{\pm0.35}$ | $14.31^{\pm0.24}$ | $177.61^{\pm0.81}$ | $48.47^{\pm0.22}$ | $55.72^{\pm0.21}$ | $34.24^{\pm0.33}$ |

CLaTr for camera) across different textual guidance values ($w_c$ in Equation 8). The optimal point lies in the bottom-right corner of each plot (low $FD_{framing}$, high modality scores). Across both architectures and modalities, our auxiliary sampling (Aux) method achieves the best performance. It highlights its effectiveness in improving both framing quality and textual alignment on both architectures and for both modalities.

**Qualitative results.** Figures 17 and 18 present qualitative results with Aux sampling on the pure subset for DiT and MAR, respectively. In these examples, the human motion is accurately aligned with the prompts: in DiT, the person jumps as specified, and in MAR, the person leans forward. In the DiT example, the camera follows the person closely both vertically and laterally, maintaining proper on-screen framing, while in both cases the camera performs smooth pull-out motions. These examples further demonstrate that Aux generates human and camera behavior that faithfully follows the input prompts, producing precise motion and well-framed sequences across architectures.

### E.2.4 ABLATION STUDY ON PURE DATASET

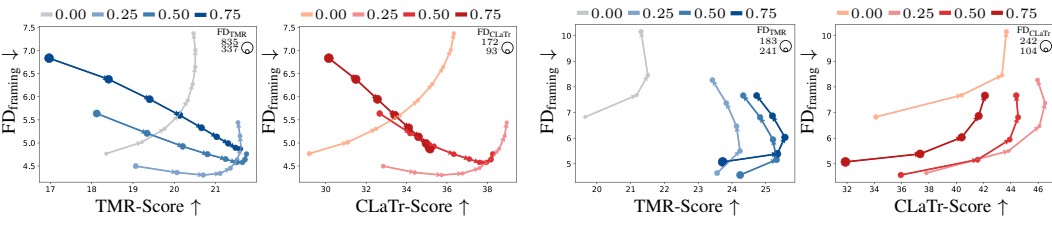

(a) FD$_{framing}$-TMR-Score  (b) FD$_{framing}$-CLaTr-Score     (a) FD$_{framing}$-TMR-Score  (b) FD$_{framing}$-CLaTr-Score

Figure 19: **Auxiliary guidance ablation in DiT.** Trade-off between framing quality and modality-text alignment for textual guidance ranges from 4 to 12. The optimal region is at the bottom-right (low framing error, high alignment).

Figure 20: **Auxiliary guidance ablation in MAR.** Trade-off between framing quality and modality-text alignment for textual guidance ranges from 1 to 5. The optimal region is at the bottom-right (low framing error, high alignment).

Similarly to the manuscript, we ablate the auxiliary guidance weight $w_z$ (Equation (8)) on both DiT and MAR on the *pure* subset; results are shown in Table 9. We see that a moderate guidance weight improves framing and text–modality alignment. On DiT, increasing $w_z$ from 0.00 to 0.25 reduces FD$_{framing}$ $6.78 \rightarrow 5.03$ and Out-rate $36.25 \rightarrow 24.92$; on MAR, $w_z=0.50$ lowers them $6.55 \rightarrow 4.90$ and $30.19 \rightarrow 24.28$.

Pushing $w_z$ further continues to aid framing but degrades fidelity: FD$_{TMR}$ and FD$_{CLaTr}$ rise (DiT $424.81 \rightarrow 460.87$, MAR $281.39 \rightarrow 308.10$).

At high weights ($w_z=0.75$ for DiT, 1.50 for MAR), the trend becomes unstable, with FD$_{TMR}$ spiking to 510.60 and FD$_{CLaTr}$ to 177.61.

We then illustrate Figures 19 and 20 for the trade-off between framing quality (FD$_{framing}$, lower is better) and text–modality alignment (TMR, CLaTr; higher is better) as the Aux guidance weight $w_z$ varies. The optimum lies near the bottom-right of each plot. Across both architectures, we see: (1) introducing guidance yields a large gain: $w_z: 0 \rightarrow 0.25$ (DiT) and 0.50 (MAR) shift points toward the bottom-right; (2) further increases, 0.50 (DiT), 1.0 (MAR), continue to improve framing but begin to reduce fidelity, reflected by larger markers (higher Fréchet distances); and (3) at very high weights, 0.75 (DiT), 1.50 (MAR), performance degrades on both axes.

### E.2.5 ABLATION STUDY ON MODALITY INDEPENDENCE

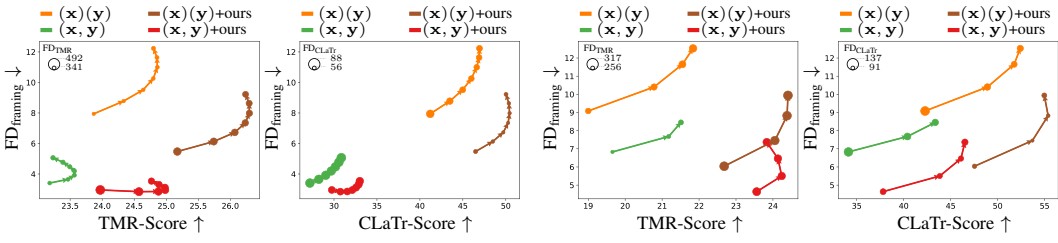

(a) FD$_{framing}$-TMR-Score  (b) FD$_{framing}$-CLaTr-Score     (a) FD$_{framing}$-TMR-Score  (b) FD$_{framing}$-CLaTr-Score

Figure 21: **Independent modality ablation in DiT on mixed subset.** Trade-off between framing quality and modality-text alignment for textual guidance ranges from 4 to 12. The optimal region is at the bottom-right (low framing error, high alignment).

Figure 22: **Independent modality ablation in MAR on mixed subset.** Trade-off between framing quality and modality-text alignment for textual guidance ranges from 4 to 12. The optimal region is at the bottom-right (low framing error, high alignment).

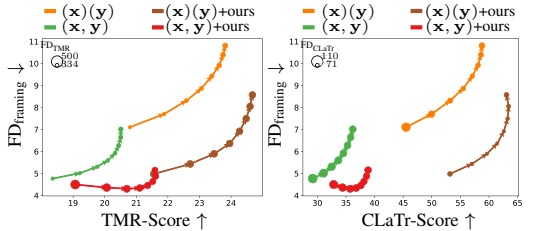 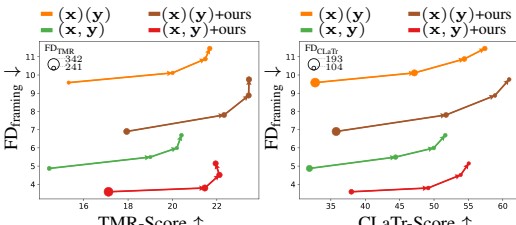

(a) FD$_{\text{framing}}$-TMR-Score  (b) FD$_{\text{framing}}$-CLaTr-Score    (a) FD$_{\text{framing}}$-TMR-Score  (b) FD$_{\text{framing}}$-CLaTr-Score

Figure 23: **Independent modality ablation in DiT on pure subset.** Trade-off between framing quality and modality-text alignment for textual guidance ranges from 4 to 12. The optimal region is at the bottom-right (low framing error, high alignment).

Figure 24: **Independent modality ablation in MAR on pure subset.** Trade-off between framing quality and modality-text alignment for textual guidance ranges from 4 to 12. The optimal region is at the bottom-right (low framing error, high alignment).

In this section, we analyze the influence of independent modality generation $(\mathbf{x}|\mathbf{y})$ versus dual-modality generation $(\mathbf{x}, \mathbf{y})$. We illustrate the trade-off between framing quality and modality-text alignment for both DiT and MAR architectures in Figure 21 and Figure 22 for the *mixed* subset, and in Figure 23 and Figure 24 for the *pure* subset.

Across all settings, the same phenomena are consistently observed:

- **The dual-modality generation setup $(\mathbf{x}, \mathbf{y})$ tends to improve inter-modality alignment at the cost of lower modality-wise performance.** This is visible in the figures when comparing green vs. orange or red vs. brown curves: the dual-modality setting appears further to the left (worse modality-wise metrics) but lower on the vertical axis (better inter-modality alignment, framing).

- **In both independent and dual-modality cases, our auxiliary sampling (Aux) consistently enhances overall performance.** Comparing green vs. red and orange vs. brown curves, Aux shifts the points toward the bottom-right, closer to the optimal balance between framing quality and modality-text alignment.

### E.2.6    QUALITATIVE VISUALIZATION OF AUXILIARY SAMPLING INFLUENCE

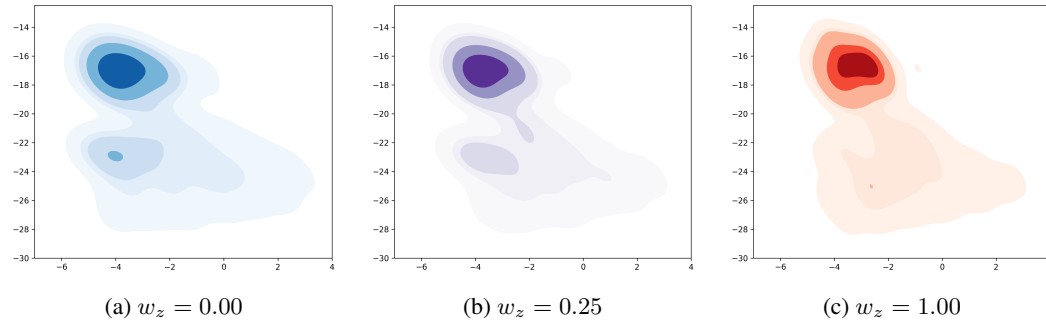

(a) $w_z = 0.00$               (b) $w_z = 0.25$               (c) $w_z = 1.00$

Figure 25: Visualization of UMAP projection density of $2,000$ generated samples for different values of auxiliary sampling weight $w_z$.

Figure 25 illustrates the evolution of UMAP projection density for a subset of generated samples as we vary the auxiliary sampling weight $w_z$. We analyze three settings: $w_z = 0.00$ in Figure 25a (no auxiliary sampling, corresponding to the $(\mathbf{x}, \mathbf{y})$ baseline); $w_z = 0.25$ in Figure 25b (the optimal value used for numerical comparisons); and $w_z = 1.00$ in Figure 25c (an extreme value chosen to exaggerate the behavior).

Overall, we observe that auxiliary sampling shifts the probability mass towards a single mode. While Figure 25a exhibits bimodality with peaks near (-4,-22.5) and (-4,-16), increasing the weight to $w_z = 1.00$ concentrates the density into a single mode around (-4,-16).

We posit that this behavior mirrors the distribution tilting effect observed in guidance-based generation (Karras et al., 2024), where stronger guidance pushes samples toward high-density regions of the conditional distribution, improving prompt adherence at the cost of reduced diversity. In our case, auxiliary sampling similarly strengthens cross-modal consistency (i.e. enhancing $FD_{framing}$ in Table 8) while diminishing sample diversity (i.e. decreasing the modality coverage metrics Table 8).

**Input prompt:**

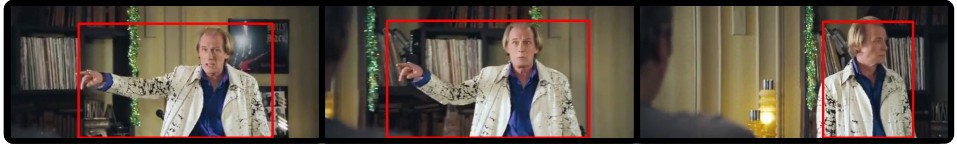

```
Task:
  Observe the full motion sequence of the highlighted person carefully.
  Summarize their motion in one concise, accurate, and complete sentence
  in English.
Instructions:
  – Clearly state the primary global actions (e.g., walk, run, jump,
  sit, squat).
  – Specify relevant details of the motion:
  – Directions (e.g., forward, backward, left, right, clockwise).
  – Body parts involved explicitly (e.g., left arm, right foot, hips).
  – Movement trajectory patterns if present (e.g., circular, zig-zag,
  straight line).
  – Movement speed or intensity (e.g., slowly, quickly, rapidly).
  – Order and sequencing of actions clearly (what happens first,
  next, last).
  – Always describe global movement explicitly, even when emphasizing
  local actions (e.g., "A person stands still and turns their head"
  instead of just "turns head").
  – Refer only to "a person," avoiding mention of highlighting or boxes.
  Avoid:
  – Inferring emotional states or intentions.
  – Vague or overly generalized descriptions ("A person moves,"
  "A person acts").
  – Excessive spatial precision or measurement details ("moves
  exactly 5 cm forward").
  Good Examples:
  – "A person rises from sitting, walks quickly in a clockwise circle,
  then sits down again."
  – "A person jumps repeatedly side-to-side while holding their
  left wrist with the right hand, then runs rapidly backward."
  – "A person crouches, then climbs upward onto an object
  positioned to their right."
  Bad Examples (DO NOT PRODUCE):
  – Over-general or no main motion: "A person looks."
  – Too detailed spatially: "A person moves exactly
  10 centimeters left."
  – Explicit mention of highlighting: "The character in the red box jumps."
  Your response must strictly follow the format of the good examples provided.
```

**Response:**

```
A person points forward with their right hand while walking slightly to the left,
then stops and turns their head to the right.
```

Figure 26: **Example of human motion caption instruction.**

