# OpenReview forum: "Pulp Motion: Framing-aware multimodal camera and human motion generation"
_ICLR.cc/2026/Conference — ICLR 2026 Poster_

### Official Review · Reviewer_XH8N · 2025-10-30

**Soundness:** 3
**Presentation:** 3
**Contribution:** 4
**Rating:** 6
**Confidence:** 5

**Summary:**

This paper addresses the problem of jointly generating human motion and camera trajectories from textual descriptions. The authors argue that treating these modalities separately often leads to poor cinematic results (e.g., the camera losing track of the actor). They propose a novel framework that uses "on-screen framing" (the 2D projection of human joints onto the camera plane) as an auxiliary modality to bridge the gap between the two. They also contribute "PulpMotion," a large-scale dataset with paired camera-human motions and rich captions for both.

**Strengths:**

- The core idea of using on-screen 2D framing as the "bridge" between 3D camera and 3D human motion is cinematographically sound.
- I appreciate that the authors tested this on two different state-of-the-art architectures (DiT and MAR). This convincingly demonstrates that their sampling method is model-agnostic and broadly applicable
- The PulpMotion dataset seems like a significant step up,, particularly with the motion refinement steps to fix occlusions and the inclusion of detailed VLM-generated captions for both modalities.

**Weaknesses:**

-  The method relies heavily on the linear transform W to map from the joint (x,y) latent space to the framing latent z. While the results are good, I wonder if this linear assumption is too restrictive for highly complex camera moves or unusual perspectives where the relationship between 3D positions and 2D projections might be highly non-linear, even in a latent space.

- The evaluation heavily relies on FD metric in various feature spaces. While standard, these can sometimes be disconnected from perceived quality. Qualitative video examples are necessary to fully trust that the "refined" motions in the new dataset don't contain artifacts from the diffusion inpainting.

- Quantitative results: Authors present metrics only for their own trained models. There is no comparison against other baselines/SOTA works that:
(1) Create human motion given a camera trajectory - Report motion metrics and in-frame joints here.
(2) Create camera trajectories given a human motion - Report camera trajectories metrics and in-frame joints here.

- Videos are required: A motion generation paper is highly expected to also present videos showing its results. I expect the authors to submit:
(1) Videos that show the steps of MotionPulp dataset generation, including the 3D pose estimation and camera trajectories + dataset post-processing to reach the final result.
(2) Videos of the generated results from the trained model.
(3) Comparison against other SOTA methods.

**Questions:**

1. Did you experiment with non-linear mappings for W? I am curious if a simple MLP would provide better framing guidance than the linear transform, or if it would break the elegant decomposition used in the sampling derivation.

2. Regarding the dataset refinement: When you use diffusion inpainting to fix out-of-frame body parts, how often does this result in plausible but essentially "hallucinated" motions that might not match the original real-world action?

3. "Since the final sampling does not explicitly condition on z, there is no need to include the bridging modality during training. This reduces training cost and yields a more general approach." (L264) - If this is your conclusion, why W and Dec_z are needed? You can drop those from your architecture and paper.

4. Are you planning to release the PulpMotion dataset and the model for reproducibility?

---

> ### Author Response · Authors · 2025-11-21
> **Response to reviewer XH8N (1/2)**
>
> Dear reviewer **XH8N**,
> Thank you for your comment and for highlighting the pertinence of using the auxiliary modality, the model-agnostic nature of our approach, and the relevance of our dataset.
>
> ### W3: Comparison against other unimodal baselines/SOTA.
>
> First, recall that we compare against **(x)(y)** that serves as a unimodal baseline, i.e., two independent models: text-to-camera and text-to-human.
> As shown in Tables 4 and 8, Figures 4, 5, 15, and 16, and in Section E.2.5, this baseline performs well on per-modality metrics (i.e., it achieves low Frechet Distance for both camera and human motion). However, it performs poorly on inter-modality metrics (i.e., the framing Frechet Distance and out-rate), as expected.
>
> As requested also by reviewer **akEG** (**W1.2**), we added **DIRECTOR** (Courant et al., 2024), a human+text-to-camera baseline, which is to our knowledge the only existing method combining both camera and human modalities.
> We report results for **DIRECTOR+(y)** (DiT on the pure subset, consistent with Table 8), where humans are first generated using the same **(y)** backbone as in the independent baseline (**(x)(y)**), after which the camera generation in **DIRECTOR** is conditioned on these synthesized humans.
> We observe that in terms of framing performance, **DIRECTOR+(y)** falls between the independent baseline (**(x)(y)**) and the joint baseline (**(xy)**), with our auxiliary sampling still outperforming them. This shows that conditioning on the human trajectory alone is insufficient to achieve strong framing.
>
> | Method              | FD$_\text{framing}$$\downarrow$ | Out-rate$\downarrow$ | FD$_\text{CLaTr}$$\downarrow$ | CLaTr-Score$\uparrow$ | CLaTr-Coverage$\uparrow$ |
> |:------------------- | -------------------------------:| --------------------:| -----------------------------:| ---------------------:| ------------------------:|
> | **(x)(y)**          |   41.70 |    10.46 |               86.06 |                 57.74 |                    31.83 |
> | **(xy)**            |    36.25 |                12.73 |                         93.37 |                 35.99 |                    44.56 |
> | **DIRECTOR+(y)**    |                            23.5 |                67.22 |                        128.72 |                 34.71 |                    41.13 |
> | **(xy)+Aux (ours)** |                            5.03 |                24.92 |                         91.36 |                 38.42 |                    40.94 |
>
> ### W4: Qualitative results.
>
> As requested (also by the other reviewers), in addition to figures in the manuscript (Figures 1, 6, 7, 10, 13, 14, 17 18), we now provide an [online gallery](https://pulp-motion.github.io/pulp-motion-gallery/) with additional qualitative videos, including generated samples, comparisons to baselines, dataset refinement, and representative dataset samples:
> - **Generation examples:** We compare outputs generated with our method to those produced by other SOTA/baselines for both architectures (DiT and MAR). Overall, our method achieves more consistent and stable framing, keeping the human on screen throughout the shot. In contrast, other methods either ignore the human entirely (**(x)(y)**) or begin to follow the subject but fail to maintain framing across the sequence (**(x,y)**, **(x,y,z)** and **ReDi**) as in the first example of the gallery.
> - **Dataset pipeline:** We show the full dataset pipeline, including extraction of camera, human, and textual information from raw video clips, as well as the refinement step. Recall that the refinement step consists of applying generative inpainting to out-of-frame body parts, where the initial estimation may be unreliable. The refinement significantly improves motion naturalness, for example, converting ground-sliding into realistic walking behavior.
> - **Dataset samples:** We also provide representative dataset samples highlighting the diversity of camera motions, human motions, and textual captions.
>
> We have added this gallery link to the supplementary material (Section E.2.2, highlighted in blue).
>
> ### Q1: Non-linear mappings for $\mathbf{W}$ for better framing guidance.
>
> All our equation derivations rely on the linearity properties of Gaussian distributions (cf. Eq. 10). Because an MLP introduces non-linear activations, the Gaussian assumption no longer holds, and the derivations become invalid under such a non-linear mapping.
> One could, in principle, extend Eq. 1 with an additional bias term, i.e., $\mathbf{z} = \mathbf{W} [\mathbf{x}, \mathbf{y}]^\top + \mathbf{b}$.
> However, this makes the derivations substantially more cumbersome, while in practice we observe that including this bias term in the *autoencoder* does **not** improve reconstruction performance of the on-screen framing.
> Importantly, since this transformation is applied in the *latent* space, the restriction to a linear mapping is not severe, non-linearities would have a much stronger effect if applied directly in the raw modality space.

---

> > ### Author Response · Authors · 2025-11-21
> > **Response to reviewer XH8N (2/2)**
> >
> > ### Q2: Hallucination during dataset refinement.
> >
> > It is very difficult to quantify this effect precisely.
> > We provide an example in Figure 10 and the corresponding video in the [online gallery](https://pulp-motion.github.io/pulp-motion-gallery/). In this example the subject is running, but because their legs are out of the screen, the 3D pose estimator incorrectly predicts a sliding motion. By combining the VLM caption with diffusion-based inpainting, we are able to synthesize a *plausible* walking motion instead.
> > Yet, because the out-of-frame joints are not observable in the raw video, there is always intrinsic ambiguity.
> >
> > Importantly, all visible joints remain untouched during this process, so any potential hallucination is strictly limited to the out-of-screen joints.
> >
> > ### Q3: Need of $\mathbf{W}$ and its decoder.
> >
> > The autoencoder is trained independently of the generation backbone. We train a $\mathbf{z}$ decoder (cf. Fig. 2) to supervise the learning of $\mathbf{W}$ through the reconstruction loss, as explained in Section 3.1.
> >
> > During sampling (Eq. 8), we require $\mathbf{P}_{\parallel}$, which depends on $\mathbf{W}$. As defined in Section C.2, $\mathbf{P} = \mathbf{W} (\mathbf{W}^T \mathbf{W})^{-1} \mathbf{W}^T$.  Consequently, $\mathbf{W}$ plays a crucial role in our auxiliary sampling.
> >
> > We agree that the connection between $\mathbf{W}$ and $\mathbf{P}_{\Vert}$  was unclear, and we have added the expression in Section 3.2 of the manuscript to clarify this link (blue color).
> >
> > ### Q4: Release of code, model, and dataset.
> >
> > Yes, all code, dataset, and model checkpoints will be made publicly available for reproducibility.

---

> ### Author Response · Authors · 2025-12-03
> **Summary of discussion with reviewer XH8N**
>
> Dear Reviewer XH8N,
>
> As the discussion period closes, we thank you again for recognizing the idea behind our auxiliary sampling, the model-agnostic nature of our approach, and the significance of our proposed dataset.
>
> Regarding your concerns, we believe the rebuttal has addressed them as follows:
>
> - **Lack of qualitative results:** We provided a [link to an online gallery](https://pulp-motion.github.io/pulp-motion-gallery/) featuring video comparisons, dataset pipeline creation and dataset samples to aid quality judgment.
> - **Unimodal/SOTA baselines:** We clarified that the **(x)(y)** baseline reported in the original manuscript represents a strong unimodal baseline. We also added new results for **DIRECTOR+(y)** (Courant et al., 2024), showing that conditioning alone is insufficient and auxiliary sampling provides superior framing.
> - **Non-linear mappings:** We explained that our auxiliary sampling derivation relies on the linearity of the Gaussian distribution, so introducing the non-linearity of an MLP would break the derivation; affine variants add complexity without improving performance, and linearity is less restrictive in latent space.
> - **Dataset refinement hallucinations:** We recalled that visible joints remain untouched, limiting possible hallucinations to unobserved regions; additional qualitative videos were provided in the gallery.
> - **Need for W and decoder:** Although we do not output $\mathbf{z}$, $\mathbf{W}$ is required to compute the projection matrix $\mathbf{P}_{//} = \mathbf{W} (\mathbf{W}^\top \mathbf{W})^{-1} \mathbf{W}^\top$. We now added this definition to Section 3.2 for better clarity.
> - **Release details:** We confirmed public release of the dataset, code, and checkpoints.
>
> Thank you for your time and consideration.

---

### Official Review · Reviewer_JSGB · 2025-10-30

**Soundness:** 1
**Presentation:** 3
**Contribution:** 3
**Rating:** 2
**Confidence:** 2

**Summary:**

The paper addresses an interesting problem of generating human motion in conjunction with camera trajectories. The paper describes a method for this generation, along with a dataset to be published for training and testing. While this all sounds interesting, there is no material to demonstrate the quality of the data, the method, or the generated results, and hence evaluating the quality of the idea and its implementation is unfortunately impossible at this stage it seems.

**Strengths:**

Ideas are good. There seems to be a large effort put into this work.

**Weaknesses:**

There are some questions regarding the implementation. For example, how well does the inpainting motion idea of occluded footage looks like, and what is the overall quality of the data in terms of foot skating etc.
There is not way to evaluated the quality unfortunately

**Questions:**

Am I missing something?

---

> ### Author Response · Authors · 2025-11-21
> **Response to reviewer JSGB**
>
> Dear reviewer **JSGB**,
> Thank you for your comment and for highlighting the good ideas and the efforts put into this work.
>
> Note that, as pointed out by reviewer **XH8N**, our code, models, and data will be made publicly available.
>
>
> ### W1: Inpainting performance.
>
> **Additional qualitative results.**
>
> As requested (also by the other reviewers), in addition to figures in the manuscript (Figures 1, 6, 7, 10, 13, 14, 17 18), we now provide an [online gallery](https://pulp-motion.github.io/pulp-motion-gallery/) with additional qualitative videos, including generated samples, comparisons to baselines, dataset refinement, and representative dataset samples:
> - **Generation examples:** We compare outputs generated with our method to those produced by other SOTA/baselines for both architectures (DiT and MAR). Overall, our method achieves more consistent and stable framing, keeping the human on screen throughout the shot. In contrast, other methods either ignore the human entirely (**(x)(y)**) or begin to follow the subject but fail to maintain framing across the sequence (**(x,y)**, **(x,y,z)** and **ReDi**) as in the first example of the gallery.
> - **Dataset pipeline:** We show the full dataset pipeline, including extraction of camera, human, and textual information from raw video clips, as well as the refinement step. Recall that the refinement step consists of applying generative inpainting to out-of-frame body parts, where the initial estimation may be unreliable. The refinement significantly improves motion naturalness, for example, converting ground-sliding into realistic walking behavior.
> - **Dataset samples:** We also provide representative dataset samples highlighting the diversity of camera motions, human motions, and textual captions.
>
> We have added this gallery link to the supplementary material (Section E.2.2, highlighted in blue).
>
> **Analysis of dataset refinement.**
>
> We recall that we reported quantitative results in Tables 2 and 3 (text-motion alignment and motion quality) and qualitative results in Figure 10 to show the effect of dataset refinement, and we compare to other approaches (i.e. motion-to-text). In addition, we now provide additional examples and videos in the [online gallery](https://pulp-motion.github.io/pulp-motion-gallery/).
>
> In addition, to quantify the improvement (also requested by reviewer **akEG**, point **W2**), we conducted an experiment analyzing walking motion before and after refinement. Specifically, we **selected 2,139 samples** whose human captions contained the word **“walking”** and analyzed the displacement of foot joints to determine whether the human is walking or not. For this, we calculated the vertical difference between the feet over time and identified **lift events**, i.e. transitions from “not lifted” to “lifted”, to count the number of steps per second. We observed:
> - **Before refinement**, 25% of samples perform at least one step every 2 seconds, and only 8% achieve at least one step per second. This implies that 75% are slower than one step every 2 seconds, which is neither realistic nor natural.
> - **After refinement:** 72% of samples perform at least 1 step every 2 seconds, or 37% perform at least 1 step per second.
>
> These results demonstrate a significant improvement in the naturalness and frequency of walking motion because the refined model generates gait patterns that more closely resemble natural locomotion (i.e. the typical human walking cadence of approximately one step per second).

---

> ### Author Response · Authors · 2025-12-03
> **Summary of discussion with reviewer JSGB**
>
> Dear Reviewer JSGB,
>
> As the discussion period closes, we thank you again for recognizing the idea behind our auxiliary sampling as well as the efforts put in this work.
>
> Regarding your concerns, we believe the rebuttal has addressed them as follows:
> - **Lack of qualitative results:** We provided a [link to an online gallery](https://pulp-motion.github.io/pulp-motion-gallery/) featuring video comparisons, dataset pipeline creation and dataset samples to aid quality judgment.
> - **Inpainting performance:** We recalled that we have reported quantitative and qualitative evidence in the manuscript; clarified that visible joints remain untouched; and added a walking-frequency analysis showing realism improves from 25% to 72% before and after the refinement step.
>
> Thank you for your time and consideration.

---

### Official Review · Reviewer_akEG · 2025-11-01

**Soundness:** 3
**Presentation:** 3
**Contribution:** 3
**Rating:** 6
**Confidence:** 3

**Summary:**

This paper presents two main contributions.
First, it proposes an auxiliary projection (Aux) method that enhances cross-modal alignment in joint human–camera generation. The approach introduces a mathematically motivated projection mechanism that can be plugged into existing diffusion frameworks to improve multi-modal consistency without architectural changes.
Second, the paper introduces PulpMotion, a large-scale dataset containing paired 3D human motions, camera trajectories, and textual captions. Compared with previous datasets such as E.T., PulpMotion offers greater scale, richer modality coverage, and improved motion quality, providing a valuable resource for future research.
Experiments demonstrate that the proposed method achieves consistent performance gains across multiple metrics and subsets.

**Strengths:**

1. The proposed auxiliary projection method is novel and well-motivated, offering a clear mathematical way to improve cross-modal alignment by decomposing the latent space and re-projecting noise predictions, and it could be applied to other multi-modal diffusion frameworks.
2. The method shows consistent, measurable improvements across metrics like framing accuracy, camera-text alignment, and motion consistency, with robust gains on both pure and mixed subsets.
3. The PulpMotion dataset is large and high-quality, combining 3D human motion, camera trajectories, and detailed text captions, roughly twice the size of previous datasets, which is  valuable for future multi-modal and generative research.

**Weaknesses:**

1. Lack of qualitative visualization, analytical ablations and comparison. (1) The paper presents extensive quantitative results but lacks visual comparisons of generated outputs across baselines like ReDi and dual-modality generation (x,y). Without qualitative visualization, it is difficult to judge whether the reported improvements in metrics correspond to genuinely better visual framing or motion quality. (2) While the paper compares against multi-modal frameworks such as ReDi and its own variants, it does not evaluate performance against established single-modality models like text-to-motion or camera-only baselines. Even if joint models might not surpass unimodal ones on single-modality metrics, such comparisons would make the results more convincing and provide clearer explanations of the trade-offs between alignment and fidelity in multi-modal generation. (3) Although the auxiliary projection P_{//}​ is mathematically well-motivated, the paper provides little empirical insight into how it improves framing or cross-modal alignment. The ablations only vary sampling weights (w_c, w_z​) and modality setups, without deeper analysis of the mechanism. Including a simple latent-space visualization (e.g., t-SNE before and after applying Aux) would help illustrate what representation changes the auxiliary term enforces. At present, the improvement appears empirical rather than well-explained.
2. The dataset itself includes human motions synthesized or refined by a diffusion-based editing model (Fig. 10). However, there is no quantitative or qualitative validation of the realism or accuracy of these generated motions. Since they serve as ground-truth for training and evaluation, this raises concerns about data fidelity.

**Questions:**

1. Can the proposed auxiliary sampling method be applied to ReDi, and would it bring similar improvements?
2. What exactly do the “pure” and “mixed” subsets in PulpMotion represent, and how are they separated?
3. What are the sources of the raw videos in PulpMotion, and were any filtering or selection criteria applied during dataset construction?

---

> ### Author Response · Authors · 2025-11-21
> **Response to reviewer akEG (1/2)**
>
> Dear reviewer **akEG**,
>
> Thank you for your comment and for highlighting the novelty, framework, and generalization of our auxiliary modality, the consistent improvements of our method across metrics, and the value of our dataset for the field.
>
> Note that, as pointed out by reviewer **XH8N**, our code, models, and data will be made publicly available.
>
>
> ### W1.1: Qualitative results.
>
> As requested (also by the other reviewers), in addition to figures in the manuscript (Figures 1, 6, 7, 10, 13, 14, 17 18), we now provide an [online gallery](https://pulp-motion.github.io/pulp-motion-gallery/) with additional qualitative videos, including generated samples, comparisons to baselines, dataset refinement, and representative dataset samples:
> - **Generation examples:** We compare outputs generated with our method to those produced by other SOTA/baselines for both architectures (DiT and MAR). Overall, our method achieves more consistent and stable framing, keeping the human on screen throughout the shot. In contrast, other methods either ignore the human entirely (**(x)(y)**) or begin to follow the subject but fail to maintain framing across the sequence (**(x,y)**, **(x,y,z)** and **ReDi**) as in the first example of the gallery.
> - **Dataset pipeline:** We show the full dataset pipeline, including extraction of camera, human, and textual information from raw video clips, as well as the refinement step. Recall that the refinement step consists of applying generative inpainting to out-of-frame body parts, where the initial estimation may be unreliable. The refinement significantly improves motion naturalness, for example, converting ground-sliding into realistic walking behavior.
> - **Dataset samples:** We also provide representative dataset samples highlighting the diversity of camera motions, human motions, and textual captions.
>
> We have added this gallery link to the supplementary material (Section E.2.2, highlighted in blue).
>
>
> ### W1.2: Comparison against other unimodal baselines/SOTA.
>
> First, recall that we compare against **(x)(y)** that serves as a unimodal baseline, i.e., two independent models: text-to-camera and text-to-human. As shown in Tables 4 and 8, Figures 4, 5, 15, and 16, and in Section E.2.5, this baseline performs well on per-modality metrics (i.e., it achieves low Frechet Distance for both camera and human motion). However, it performs poorly on inter-modality metrics (i.e., the framing Frechet Distance and out-rate), as expected.
>
> As requested also by reviewer **XH8N** (**W3**), we added **DIRECTOR** (Courant et al., 2024), a human+text-to-camera baseline, which is to our knowledge the only existing method combining the both camera and human modalities.
> We report results for **DIRECTOR+(y)** (DiT on the pure subset, consistent with Table 8), where humans are first generated using the same **(y)** backbone as in the independent baseline (**(x)(y)**), after which the camera generation in **DIRECTOR** is conditioned on these synthesized humans.
> We observe that in terms of framing performance, **DIRECTOR+(y)** falls between the independent baseline (**(x)(y)**) and the joint baseline (**(xy)**), with our auxiliary sampling still outperforming them. This shows that conditioning on the human trajectory alone is insufficient to achieve strong framing.
>
>
> | Method                | FD$_\text{framing}$$\downarrow$ | Out-rate$\downarrow$ | FD$_\text{CLaTr}$$\downarrow$ | CLaTr-Score$\uparrow$ | CLaTr-Coverage$\uparrow$ |
> |:--------------------- | -------------------:| --------:| -----------------:| -----------:| --------------:|
> | **(x)(y)**            |               41.70 |    10.46 |             86.06 |       57.74 |          31.83 |
> | **(xy)**              |               36.25 |    12.73 |             93.37 |       35.99 |          44.56 |
> | **DIRECTOR+(y)**      |                23.5 |    67.22 |            128.72 |       34.71 |          41.13 |
> | **(xy)+Aux (ours)**   |                5.03 |    24.92 |             91.36 |       38.42 |          40.94 |
>
> ### W1.3: Latent-space visualization.
>
> Thanks for pointing this out, we agree that a qualitative analysis of the effect of auxiliary sampling helps build intuition.
>
> We have added such an analysis in Section E.2.6 of the paper (blue color), including plots showing how the UMAP density of generated samples evolves as the auxiliary sampling weight $w_z$ increases.
> As discussed there, larger $w_z$ values progressively concentrate the distribution into a single dominant mode, exhibiting a guidance-like tilting effect similar to that described in Karras et al. (2024): auxiliary sampling strengthens cross-modal consistency (e.g., improved framing) while slightly reducing sample diversity.

---

> > ### Author Response · Authors · 2025-11-21
> > **Response to reviewer akEG (2/2)**
> >
> > ### W2: Quantitative and qualitative evaluation of dataset realism.
> >
> > We recall that we reported quantitative results in Tables 2 and 3 (text-motion alignment and motion quality) and qualitative results in Figure 10 to show the effect of dataset refinement, and we compare to other approaches (i.e. motion-to-text). In addition, we now provide additional examples and videos in the [online gallery](https://pulp-motion.github.io/pulp-motion-gallery/).
> >
> > In addition, to quantify the improvement (also requested by reviewer **JSGB**, point **W1**), we conducted an experiment analyzing walking motion before and after refinement. Specifically, we **selected 2,139 samples** whose human captions contained the word **“walking”** and analyzed the displacement of foot joints to determine whether the human is walking or not. For this, we calculated the vertical difference between the feet over time and identified **lift events**, i.e. transitions from “not lifted” to “lifted”, to count the number of steps per second. We observed:
> > - **Before refinement**, 25% of samples perform at least one step every 2 seconds, and only 8% achieve at least one step per second. This implies that 75% are slower than one step every 2 seconds, which is neither realistic nor natural.
> > - **After refinement:** 72% of samples perform at least 1 step every 2 seconds, or 37% perform at least 1 step per second.
> >
> > These results demonstrate a significant improvement in the naturalness and frequency of walking motion because the refined model generates gait patterns that more closely resemble natural locomotion (i.e. the typical human walking cadence of approximately one step per second).
> >
> >
> > ### Q1: Auxiliary sampling with ReDi.
> >
> > **Training cost and modality requirements.**
> > Our auxiliary sampling approach achieves cross-modal improvements while training on only two modalities $(\mathbf{x},\mathbf{y})$, making it both efficient and easy to deploy. In contrast, **ReDi requires training on all three modalities** $(\mathbf{x},\mathbf{y},\mathbf{z})$; applying auxiliary sampling on top of ReDi would remove this advantage.
> >
> > **Redundancy of information.**
> > The projection term $\mathbf{P}_{//}$ in our auxiliary sampling (see Section C.3) already encodes the information contained in $\mathbf{z}$. Because of this, we expect limited additional benefit from explicitly injecting $\mathbf{z}$ again via ReDi.
> >
> > **Empirical results confirm this.**
> > We performed an explicit comparison using DiT on the pure subset (consistent with Table 8). The results are reported below.
> > We observe that **(xyz)+ReDi+Aux** performs similarly or worse than **(xyz)+Aux**. Comparing **(xyz)+ReDi**, **(xyz)+Aux**, and **(xyz)+ReDi+Aux** shows that the improvements primarily come from our auxiliary sampling rather than from ReDi.
> >
> > | Method             | FD$_\text{framing}$$\downarrow$ | Out-rate$\downarrow$ | FD$_\text{TMR}$$\downarrow$ | TMR-Score$\uparrow$ | TMR-Coverage$\uparrow$ | FD$_\text{CLaTr}$$\downarrow$ | CLaTr-Score$\downarrow$ | CLaTr-Coverage$\uparrow$ |
> > |:------------------ | -------------------:| --------:| ---------------:| ---------:| ------------:| -----------------:| -----------:| --------------:|
> > | **(xyz)**          |                5.56 |    29.81 |          334.29 |     18.04 |        15.52 |            108.05 |       28.62 |          45.83 |
> > | **(xyz)+ReDi**     |                6.53 |    33.34 |          323.53 |     17.13 |        15.21 |             99.60 |       28.65 |          48.60 |
> > | **(xyz)+Aux**      |                4.63 |    22.98 |          435.40 |     19.42 |        15.10 |             83.07 |       31.79 |          47.65 |
> > | **(xyz)+ReDi+Aux** |                4.93 |    21.56 |          483.03 |     19.20 |        11.86 |             86.48 |       30.67 |          46.00 |
> >
> >
> >
> > ### Q2: *Pure* and *mixed* subsets.
> >
> > We split the dataset into "*pure*” and "*mixed*” subsets, similar to the E.T. dataset (Courant et al., 2024):
> > - **Pure subset:** Contains only samples with a single camera motion trajectory (e.g. *“the camera trucks right”*); smaller and simpler.
> > - **Mixed subset:** Contains all the samples, excluding some static-only camera motion samples to make it more balanced; larger but complex than the pure subset.
> >
> > ### Q3: Sources of raw videos in PulpMotion and postprocessing details.
> >
> > Thank you for pointing this out. We updated Sections 4.2 and D.1 of the manuscript to include these details.
> >
> > - **Source:** PulpMotion is built on the CondensedMovies (Bain et al., 2020) video dataset.
> > - **Postprocessing:** We follow the same extraction and postprocessing steps as E.T. (Courant et al., 2024): splitting videos into shots to obtain continuous sequences, filtering out too short shots, and extracting camera and human 3D poses. We then postprocess the extracted camera and human poses by removing velocity outliers and applying smoothing.

---

> ### Author Response · Authors · 2025-12-03
> **Summary of discussion with reviewer akEG**
>
> Dear Reviewer akEG,
>
> As the discussion period closes, we thank you again for recognizing the novelty, mathematical soundness and effectivness of our auxiliary sampling as well as the relevance of our proposed dataset.
>
> Regarding your concerns, we believe the rebuttal has addressed them as follows:
> - **Lack of qualitative results:** We provided a [link to an online gallery](https://pulp-motion.github.io/pulp-motion-gallery/) featuring video comparisons, dataset pipeline creation and dataset samples to aid quality judgment.
> - **Unimodal/SOTA baselines:** We clarified that the **(x)(y)** baseline reported in the original manuscript represents a strong unimodal baseline. We also added new results for **DIRECTOR+(y)** (Courant et al., 2024), showing that conditioning alone is not sufficient and auxiliary sampling provides better framing.
> - **Auxiliary-sampling analysis**: We included new UMAP density plots (Section E.2.6), confirming that auxiliary sampling concentrates the distribution and effectively guides generation similar to what is described in Karras et al. (2024).
> - **Dataset realism:** We recalled that we have already reported quantitative and qualitative evidence in the manuscript; clarified that visible joints remain untouched; and added a walking-frequency analysis showing realism improves from 25% to 72% before and after refinement step.
> - **Auxiliary sampling with ReDi**: We added a new experiment combining our auxiliary sampling with ReDi, yielding marginal gains, as our projection term already captures the useful signal; our method is also more flexible since it does not require to generate $\mathbf{z}$.
> - **Clarifications:** We updated Section 4.2 and D.1 to detail *pure/mixed* subsets (as in Courant et al., 2024) and the source of raw videos (CondensedMovies).
>
> Thank you for your time and consideration.

---

### Official Review · Reviewer_eHvw · 2025-11-03

**Soundness:** 2
**Presentation:** 2
**Contribution:** 2
**Rating:** 4
**Confidence:** 4

**Summary:**

This work examines the problem of joint synthesis of human motions and camera motions. To train the model, the paper introduces a dataset of containing samples of human motions, camera motions, and text descriptions which are gathered from videos of human motions using SOTA pose, camera motion, and captioning models. The generation model consists of two parts: an autoencoder which encodes human motions and camera motions into a latent space, and a diffusion model which generates camera and motion latents which are decoded into the final camera and motion samples. The autoencoder also models the on-screen framing in the latent space as a linear transformation of the human and camera latents. A variant of CFG that is based on orthogonal projection on the kernel space of the framing latent linear projection is introduced to promote strong alignment with the text description. Experiments are conducted to investigate the ability of the proposed model to jointly generate human motion and camera motions given a text description.

**Strengths:**

* Joint generation of human and camera motion is an important problem which could lead to generative models that are suitable for achieving cinematic or artistic effects.
* The proposed dataset has a large number of samples covering diverse situations. The proposed dataset is the first to focus equally on the gathering and annotating both poses and camera motions.
* Experimental results show that the proposed dataset can be used to learn joint human and camera motions.

**Weaknesses:**

* The submission did not include example videos of the collected data or samples in a supplementary material file. This makes it very hard to judge the quality of the data or the sampling results. Given the lack of established metrics in this domain, human visual inspection remains the most reliable way to determine quality.
* The importance of the framing variable $z$ is unclear to me. As far as I can tell, $z_{raw}$ is never defined in the text. Furthermore, z only appears to affect the final sampling result via the encoder, and does not influence the training of the diffusion model beyond this. In the comparisons in Figure 4 and 5, including z generally reduces the quality of results compared only using x and y when auxiliary sampling (essentially a variant of CFG) is not used. I do not feel that the importance of z has been fully validated.
* I am not sure if the importance of the auxiliary sampling in equation (8) is fully validated compared to standard CFG in equation (4). I did not see an experiment comparing the effect of (8) vs. (4) given the same trained model. Am I missing this, or can it be added? Furthermore, was CFG used for the baseline models in Section 5.2? If not, the comparison with the proposed method is not fair because guidance makes a major impact on sampling results.

**Questions:**

* Can visualizations of data samples and model samples be shown for qualitative analysis?
* Why does including z make results worse in Figure 4 and 5, but better when auxiliary sampling is used?
* Can you compare auxiliary sampling to standard CFG for the same trained model?
* Was CFG used for the baseline models in Figure 4 and 5?
* What is the definition of $z_{raw}$?

---

> ### Author Response · Authors · 2025-11-21
>
> Dear reviewer **eHvw**,
> Thank you for your thoughtful comments and for highlighting the relevance of our task and dataset.
>
> Note that, as pointed out by reviewer **XH8N**, our code, models, and data will be made publicly available.
>
> ### Q1: Qualitative results.
>
> As requested (also by the other reviewers), in addition to figures in the manuscript (Figures 1, 6, 7, 10, 13, 14, 17 18), we now provide an [online gallery](https://pulp-motion.github.io/pulp-motion-gallery/) with additional qualitative videos, including generated samples, comparisons to baselines, dataset refinement, and representative dataset samples:
> - **Generation examples:** We compare outputs generated with our method to those produced by other SOTA/baselines for both architectures (DiT and MAR). Overall, our method achieves more consistent and stable framing, keeping the human on screen throughout the shot. In contrast, other methods either ignore the human entirely (**(x)(y)**) or begin to follow the subject but fail to maintain framing across the sequence (**(x,y)**, **(x,y,z)** and **ReDi**) as in the first example of the gallery.
> - **Dataset pipeline:** We show the full dataset pipeline, including extraction of camera, human, and textual information from raw video clips, as well as the refinement step. Recall that the refinement step consists of applying generative inpainting to out-of-frame body parts, where the initial estimation may be unreliable. The refinement significantly improves motion naturalness, for example, converting ground-sliding into realistic walking behavior.
> - **Dataset samples:** We also provide representative dataset samples highlighting the diversity of camera motions, human motions, and textual captions.
>
> We have added this gallery link to the supplementary material (Section E.2.2, highlighted in blue).
>
> ### Q2: Including $\mathbf{z}$ makes results worse.
>
> Thanks for pointing this out, this is indeed a key remark.
> We have two explanations:
> - Including $\mathbf{z}$ tends to degrade performance because the model must allocate more capacity to represent $\mathbf{z}$ even though it is fully deterministic (then redundant) given the fixed parameter and training budget;
> - As in multimodal or multi-task learning, introducing additional losses can lead to gradient conflicts, which makes optimization more difficult.
>
> That said, the primary goal of our paper is to demonstrate the __effectiveness of the proposed auxiliary sampling__ and its benefits.
> Your remark is therefore very relevant, and to emphasize that our contribution is not tied to a specific backbone and modality combinations, we also report results using the **(xyz)** backbone with our auxiliary sampling (**(xyz)+Aux**) on the pure dataset for DiT (consistent with Table 8).
> Even in this setting, we observe consistent improvements notably for the framing quality and out-of-screen rate.
>
> | Method             | FD$_\text{framing}$$\downarrow$ | Out-rate$\downarrow$ | FD$_\text{TMR}$$\uparrow$ | TMR-Score$\uparrow$ | TMR-Coverage$\uparrow$ | FD$_\text{CLaTr}$$\downarrow$ | CLaTr-Score$\uparrow$ | CLaTr-Coverage$\uparrow$ |
> |:------------------ | -------------------:| --------:| ---------------:| ---------:| ------------:| -----------------:| -----------:| --------------:|
> | **(xyz)**          |  5.56 |    29.81 |   334.29 |     18.04 |        15.52 |   108.05 |       28.62 |          45.83 |
> | **(xyz)+ReDi**     |  6.53 |    33.34 |  323.53 |     17.13 |        15.21 |  99.60 |       28.65 |          48.60 |
> | **(xyz)+Aux**      |  4.63 |    22.98 |  435.40 |     19.42 |        15.10 |           83.07 |       31.79 |          47.65 |
>
>
> ### Q3/4: CFG for auxiliary sampling and baselines.
>
>
> Figures 4 and 5 illustrate how the model’s scores vary with the CFG weight for text, $w_c$ (Eq. 4).
> For fair comparison, all other hyperparameters are optimal, i.e., we fix the auxiliary rate $w_z$ in our method and the representation rate in ReDi in their optimal values.
>
> The comparison between CFG-only (text-only guidance; Eq. 4) and auxiliary sampling (Eq. 8) corresponds to **(xy)** vs **(xy)+Aux** (Eq.4 is exactly Eq. 8 with $w_z = 0$).
>
> All comparisons use the same trained backbone. In total, we train only three backbones per architecture:
> - **(x)(y)**: independent backbones for each modality
> - **(xy)**: joint dual-modal backbone
> - **(xyz)**: joint triplet backbone
>
> The **Aux** results use the **(xy)** backbone, and **ReDi** uses the **(xyz)** backbone.
>
> In short, our auxiliary sampling approach is used alongside the text CFG weight to further boost cross-modal performances, e.g., yielding better framing quality.
>
> ### Q5: Definition of $\mathbf{z}_{\text{raw}}$.
>
> $z_{\text{raw}}$ defines the raw on-screen framing introduced in Section 5.1: 2D Normalized Device Coordinates (NDC), for nine key human joints (ankles, pelvis, spine, head, shoulders, wrists).
> Thank you for pointing out the ambiguity, we have clarified the definition in Section 3.1 and 5.1 (blue color).

---

> ### Author Response · Authors · 2025-12-03
> **Summary of discussion with reviewer eHvw**
>
> Dear Reviewer eHvw,
>
> As the discussion period closes, we thank you again for recognizing the importance of the problem and the relevance of the PulpMotion dataset.
>
> Regarding your concerns, we believe the rebuttal has addressed them as follows:
>
> - **Lack of qualitative results:** We provided a [link to an online gallery](https://pulp-motion.github.io/pulp-motion-gallery/) featuring video comparisons, dataset pipeline creation and dataset samples to aid quality judgment.
> - **Impact of explicit $\mathbf{z}$ generation:** We clarified that adding $\mathbf{z}$  forces the model to allocate capacity unnecessarily and could cause gradient conflicts. Finally, we added the comparison **(xyz)** vs. **(xyz)+Aux** highlighting that the effectiveness of our auxiliary sampling is backbone-independent.
> - **CFG & Baselines:** We clarified that our method works alongside standard text CFG, not as a replacement, and the paper has already included these comparisons.
> - **Clarifications:** We updated Sections 3.1 and 5.1 to clarify the definition of  $\mathbf{z}_\text{raw}$ as 2D Normalized Device Coordinates (NDC) for the projection of key human joints.
>
> Thank you for your time and consideration.

---

### Author Response · Authors · 2025-12-03
**General summary of reviews and discussion**

Dear Area Chair and Reviewers,

Thank you very much for your careful reviews, questions, and constructive feedback that led us to improve our paper by running new experiments and adding clarifications.

In this work, we introduce an auxiliary sampling method for diffusion models that enhances multimodal generation. It leverages an auxiliary modality that links the generated modalities and improves cross-modal coherence. We apply this method for text-to-human-and-camera generation and present the PulpMotion dataset to support and demonstrate the effectiveness of our method.

As a closing summary for the rebuttal period, we outline below the *main* actions taken and clarifications provided in response to the raised concerns:

- **Lack of qualitative results (eHvw, akEG, JSGB, XH8N):** We provided a [link to an online gallery](https://pulp-motion.github.io/pulp-motion-gallery/) featuring video comparisons, dataset pipeline creation and dataset samples to aid quality judgment.

- **Comparison against other unimodal baselines/SOTA (akEG, XH8N):** We clarified that the **(x)(y)** baseline reported in the original manuscript represents a strong unimodal baseline. We also added new results for **DIRECTOR+(y)** (Courant et al., 2024), showing that conditioning alone is not sufficient and auxiliary sampling provides better framing.

- **Data refinement step assessment (akEG, JSGB, XH8N):** We recalled that we report quantitative and qualitative evidence in the manuscript; clarified that visible joints remain untouched (limiting possible hallucinations to unobserved regions); and added a walking-frequency analysis showing realism improves from 25% to 72% before and after refinement step.

- **Impact of explicit $\mathbf{z}$ generation (eHvw):** We clarified that adding $\mathbf{z}$  forces the model to allocate capacity unnecessarily and could cause gradient conflicts. Finally, we added the comparison **(xyz)** vs. **(xyz)+Aux**, highlighting that the effectiveness of our auxiliary sampling is backbone-independent.

- **Latent-space visualization (akEG):** We included UMAP density plots (Section E.2.6), confirming that auxiliary sampling concentrates the distribution and effectively guides generation.

- **Minor clarifications:**
    - **(eHvw)** We have updated Sections 3.1 and 5.1 of the manuscript for better clarity on $\mathbf{z}_\text{raw}$ definition.
    - **(akEG)** We have updated Sections 4.2 and D.1 of the manuscript to clarify the *pure* and *mixed* subset definitions and the source of raw videos.
    - **(XH8N)** We confirmed that the code, model checkpoints, and the PulpMotion dataset will all be publicly released for reproducibility.
    - **(XH8N)** We added the definition of $\mathbf{P}_{//}$ in Section 3.2 to enhance the clarity.

Thank you again for your time, thoughtful feedback, and consideration.

---

### Meta-Review · Area_Chair_rfM9 · 2025-12-25

**Summary:**

The initial reviews highlighted several shared concerns: the absence of video demonstrations, a lack of comparative experiments to justify specific design choices, and a need for technical clarification. During the rebuttal phase, the authors provided video results that demonstrate notable quality and superiority. Furthermore, the rebuttal included strong empirical evidence supporting the proposed techniques and addressed most methodological questions effectively. Given the general consensus that the task is interesting and the results are robust, I recommend Accept (Poster).

**Reviewer Concerns:**

Addressed during Rebuttal:
- The lack of video examples originally made it difficult to assess data quality and sampling results. This has been fully addressed with the provided visualizations.
- The authors successfully integrated comparisons against additional unimodal baselines and state-of-the-art (SOTA) methods, maintaining a performance advantage.
- Concerns regarding the quantitative and qualitative evaluation of the dataset's realism were clarified as already illustrated in the main paper.
- The motivation for using a linear rather than a non-linear mapping for $(x, y) \to z$ was well-explained.
- While the rebuttal text was somewhat ambiguous, the data in Table 4 (comparing $(x, y)$ vs. $(x, y) + \text{aux}$) successfully validates the superiority of the proposed auxiliary Classifier-Free Guidance (CFG).

Outstanding/Minor Points:
- There was some disconnect between the reviewers' concerns and the authors' response regarding the role of $z$. However, I interpret $z$ primarily as a tool for constructing an auxiliary loss via a mapping matrix rather than a joint modeling component. Therefore, I do not view this as a significant weakness.
- One reviewer suggested more extensive evaluation metrics; while not fully exhaustive, the current evaluation is sufficient to support the paper's main contributions.

**Reviewer Scores:**

Reviewer eHvw (Rating: 4 $\to$ 5/6): The provision of video examples and dataset visualizations addresses this reviewer's primary grievance. While the responses regarding variable $z$ and auxiliary CFG could have been more precise, the overall evidence remains strong enough to warrant a score increase.

Reviewer akEG (Rating: 6 $\to$ 6): The rebuttal addressed all specific concerns, though the reviewer’s feedback suggests they view the work as solid but perhaps not high-impact enough to move to a higher score.

Reviewer JSGB (Rating: 2 $\to$ 6): This reviewer found the paper interesting and well-prepared but initially gave a low score due to the lack of qualitative video results and misundertanding some points. Having seen these results and maintaining a positive tone in discussions, a significant upgrade is expected.

Reviewer XH8N (Rating: 6 $\to$ 6): Most weaknesses were addressed. The minor remaining point regarding more extensive metrics is unlikely to result in a score decrease from the initial positive stance.

---

### Decision · Program_Chairs · 2026-01-26

Accept (Poster)